# Reconstruction of droughts in India using multiple land surface models (1951-2015)

Vimal Mishra[1], Reepal Shah[1], Syed Azhar[1], Harsh Shah[1], Parth Modi[1], Rohini Kumar[2]

[1]Civil Engineering, Indian Institute of Technology (IIT) Gandhinagar, Gujarat, 382355
[2]UFZ-Helmholtz Centre for Environmental Research, Leipzig, Germany

*Correspondence to*: Vimal Mishra (vmishra@iitgn.ac.in)

**Abstract.** India has witnessed some of the most severe droughts in the current decade and severity, frequency, and areal extent of droughts have been increasing. As a large population of India is dependent on agriculture, soil moisture drought affecting agricultural activities (crop yields) have significant impacts on socio-economic conditions. Due to limited observations, soil moisture is generally simulated using land surface hydrological models (LSMs); however, these LSM outputs have uncertainty due to many factors including errors in forcing data and model parameterization. Here we reconstruct agricultural drought events over India during the period of 1951-2015 based on simulated soil moisture from three LSMs, the Variable Infiltration Capacity (VIC), the Noah, and the Community Land Model (CLM). Based on simulations from the three LSMs, we find that major drought events occurred in 1987, 2002, and 2015 during the monsoon season (June through September). During the Rabi season (November through February), major soil moisture droughts occurred in 1966, 1973, 2001, and 2003. Soil moisture droughts estimated from the three LSMs are comparable in terms of their spatial coverage, however, differences are found in drought severity. Moreover, we find a higher uncertainty in simulated drought characteristics over a large part of India during the major crop-growing season (Rabi season, November to February: NDJF) compared to those of the monsoon season (June to September: JJAS). Furthermore, uncertainty in drought estimates is higher for severe and localized droughts. Higher uncertainty in the soil moisture droughts are largely due to the difference in model parameterizations (especially soil-depth); resulting in different persistence of soil moisture simulated by the three LSMs. Our study highlights the importance of accounting for the LSMs uncertainty and consideration of the multi-model ensemble system for the real-time monitoring and prediction of drought over India.

## 1. Introduction

Drought is among the top natural disasters that affect food and fresh water security. The 2014-2015 drought in India affected more than 3.3 million people and resulted in the loss of INR 6,50,000 crore (Indian express, May 11[th], 2016). Drought characteristics such as frequency, areal extent, and intensity of droughts have increased in India, which can be attributed to erratic summer monsoon as well as an increase in air temperature (Shah and Mishra, 2014). Moreover, the frequency of

severe and widespread multi-year droughts has also increased during the recent decades (Mishra et al., 2016). For instance, India has experienced 10 major droughts between 1950 and 1989 while five droughts occurred after 2000 (Pai et al., 2017). The drought of 2015 was among the most severe droughts during the period of 1901-2015, which caused enormous damage to crops and affected various sectors of society (Mishra et al., 2016). Precipitation deficit during the monsoon (rainy) season not only affects water availability during that season but it also affects the water availability in the post-monsoon (dry) season.

Despite an increase in irrigation infrastructure during the last few decades, about 66% of the Indian agriculture remains rain-fed and largely relying on the monsoon season rainfall, which accounts for about 80% of the total annual rainfall. Precipitation deficit during the monsoon season leads to deficit in root-zone soil moisture during the post monsoon crop-growing season. This deficit in the soil moisture can be amplified by positive temperature anomalies during the growing season. Due to lack of long-term observations of soil moisture, the impacts of climate variability and climate change on soil moisture drought are often studied using land surface (hydrologic) models (LSMs, Mishra et al., 2014; Sheffield and Wood, 2008; Samaniego et. al, 2013). However, these LSMs have differences in model parameterization and representation of hydrological process (Mishra et al., 2017; Wang et al., 2009), which can lead to uncertainty in simulated soil moisture. Moreover, soil depths specified in the LSMs vary depending on an individual model configuration, which can lead to differences in soil moisture persistence (Wang et al. 2009). Soil moisture persistence is important to understand the dynamics of soil moisture in response to meteorological forcing. For instance, LSMs with low soil moisture persistence may show higher sensitivity to temperature and/or precipitation anomalies. Differences in soil moisture persistence in LSMs can lead to uncertainty in drought monitoring and assessment.

There have been several projects on the inter-comparison of soil moisture and other hydrologic fluxes from different LSMs. For instance, the Global Land Data Assimilation System (GLDAS; Rodell et al., 2004), the earthH2Observe project (Beck et al., 2016), the Project for Intercomparison of Land Surface Parameterization Schemes (PILPS; Bowling et al., 2003), and the Global Soil Wetness Project (Dirmeyer et al., 1999, 2006) provide useful insights on the differences in soil moisture simulations from the LSMs and hydrological models. Our aim here is to understand the uncertainty in soil moisture drought characteristics simulated using three LSMs over India. We use observed gridded meteorological data to force the calibrated VIC, Noah, and CLM land surface models. We estimate drought indices based on precipitation and 60 cm depth (as surrogate for root-zone depth) soil moisture from the three LSMs to reconstruct the major drought events that occurred during the period of 1951-2015. We selected the top 60 cm to analyse soil moisture drought because, for many crops, effective root zone depth falls in this region (Jalota and Arora 2002). Also we consider this depth for all three models in our drought assessment so as to reduce uncertainty due to specification of different root-zone depth (based on respective vegetation parameters) and soil layer thickness in three LSMs. We recognize that their exists different sources of uncertainty in model estimates (of soil moisture) including that arises from errors in input variables (e.g., meteorological forcings, surface and sub-surface characteristics), however, here, our aim is not to quantify uncertainty due to all the sources rather we

limit ourselves on understanding the uncertainty in historical reconstructions of soil moisture droughts over India due to structural differences among different LSMs.

## 2. Methodology

### 2.1 Data

The three LSMs (VIC, Noah, and CLM) used in this study were forced with a common meteorological dataset that comprises daily precipitation, maximum and minimum temperatures, and wind speed. We used 0.25° daily gridded precipitation product available for the period 1901-2015 from India Meteorological Department (hereafter IMD; Pai et al. 2015), which was developed by IMD using data from 6995 gauge station across India and inverse distance weighting scheme (Shepard, 1984). In the gridded precipitation data from IMD, orographic, and topographic features of precipitation are well captured along with the spatial variability associated with the Indian summer monsoon. We used 1° daily gridded maximum and minimum air temperatures from IMD (Srivastava et al., 2009), which were developed using the data from 395 observation stations across India. We re-gridded air temperatures from 1° to 0.25° using method described in Maurer et al. 2002, which is based on temperature lapse rate of 6.5°C/km rise in elevation and the SYMAP algorithm. In re-gridding of air temperature, we used 0.25° digital elevation model (DEM) that was resampled from the original 30 m elevation data from the Shuttle Radar Topography Mission (SRTM). The gridded precipitation and air temperature products have been used in many previous studies on drought and heat waves (Shah et al., 2017a; Shah and Mishra, 2014, 2015, 2016b; Mishra et al., 2016).

### 2.2 Land Surface Models

We used simulated soil moisture from the three LSMs: the VIC, the Noah, and the CLM to assess uncertainty in root-zone (60 cm) depth soil moisture and retrospective drought assessment. These three LSMs have been widely used for producing land surface fluxes at global and regional scales (Rodell et al., 2004; Shah and Mishra, 2015; Unnikrishnan et al., 2013; see also Table S1 for brief description of major hydrological processes). All the three LSMs were forced with the same meteorological forcing from IMD at 0.25° resolution, and additional (radiation-related) forcing variables for the Noah and CLM were derived from the MTCLIM algorithm integrated in the VIC model (see Bohn et al., 2013). This will keep basic forcing data consistent across models. In all LSMs routines are there which disaggregate daily precipitation uniformly to sub-daily time scale, while temperature and radiation are temporally disaggregated following the diurnal cycle. Regarding the usage of land-surface datasets, we note here that each model used a slightly different set of input datasets. Soil textural properties and resulting hydrologic parameters (like field capacity, wilting points, and available water) are mostly derived based on the Harmonized World Soil Database (HWSD) and the Food and Agriculture Organization (FAO) based soil maps. However, since that land surface models use soil parameters from the two different sources that can lead to differences in

available water (Fig. S2), we evaluated the difference in the simulated soil moisture anomalies from the VIC model using the soil parameters based on the FAO and HWSD datasets (Fig. S3). We do not find a significant difference in the simulated soil moisture anomalies based on the soil parameters from the two sources. It is worth noting that the underlying (soil-textural) dataset for the HWSD is mostly derived based on the FAO dataset. The different products may use different pedo-transfer functions to derive soil related parameters. Nevertheless the derived (static) soil parameters do not induce significant differences in the temporal dynamics of simulated soil moisture anomalies (Fig. S2-S3).

Vegetation characteristics specified within LSMs were derived from the Advanced Very High-Resolution Radiometer (AVHRR) and Moderate Resolution Imaging Spectroradiometer (MODIS) datasets. We evaluated the sensitivity of vegetation parameters derived from AVHRR and MODIS on simulated soil moisture using Noah model (Fig. S4). In this case also we do not find any substantial differences in the dynamics of the simulated soil moisture anomalies due to differences in the underlying vegetation parameters estimated from AVHRR and MODIS datasets (Fig. S4). We note that the analysis conducted here is at a 0.25° spatial resolution – as a result the differences among the model simulations due to fine scale soil and vegetation characteristics may not be observed at a coarse scale.

All three LSMs were first spun-up using 65 years (1951-2015) data to establish initial conditions for the modelled states and fluxes. All the three LSMs were manually calibrated to match simulated monthly streamflow with observed stream flow data obtained from the India-WRIS (www.india-wris.gov.in) at the gauging stations (Shah and Mishra, 2016a) that are least affected by human interventions related to water diversion and water withdrawal for irrigation (see Supplemental Figure S1 and Table S2 for geographical location of these stations). We identified calibration parameters for each LSMs based on prior studies (Cai et al., 2014; Hogue et al., 2005; and Nijssen et al., 2001) and by performing a simple (one parameter at a time) sensitivity analysis. We used soil thickness also as calibration parameters following the success of calibrating the VIC using soil layers thickness (Nijssen et al., 2001; Shah and Mishra, 2016). The calibration parameters were manually adjusted so as to match observed streamflow (see Table S2 in Supplementary material). Further, we evaluated the model skill by comparing simulated soil moisture with station and satellite-based soil moisture, and also comparing the total column soil moisture changes with terrestrial water change derived based on GRACE products (see section 3.1 for more details).

### 2.2.1 The Variable Infiltration Capacity (VIC) Model

We used the VIC v4.2.a (Liang et al., 1994), which is a semi-distributed, physically based hydrologic model in water balance mode at daily time step. The VIC model simulates water and energy fluxes in each grid cell considering soil and vegetation parameters, and meteorological forcing as input. The model estimates total evapotranspiration as a sum of the canopy and bare soil evaporation and transpiration from vegetation mosaics. Any number of vegetation types can be represented within a

grid cell to represent sub-grid variability in vegetation cover. Infiltration is estimated using a variable infiltration capacity curve. The VIC model has three soil layers and the top two layers respond quickly to rainfall, and diffusion is allowed from the middle to top layers when the middle (second) layer is wet. Base flow from the bottom (third) layer is estimated using the Arno model formulation (Franchini and Pacciani, 1991). The bottom layer responds slowly to depict seasonal soil moisture

behaviour. We calibrated the VIC model parameters that include: depth of the soil layers, infiltration curve parameters, and parameters related to base-flow following (Nijssen et al., 2001). Vegetation and soil texture used in the VIC model were developed using the 1km Advanced Very High-Resolution Radiometer (AVHRR) and Harmonized World Soil Database (HWSD), respectively as described in Table S1. The VIC  model requires soil parameters like field capacity and wilting point, which were derived by first identifying soil class based on United States Department of Agriculture (USDA)

classification and then applying the Pedo-Transfer functions of Cosby et al., 1984. Soil texture specified at 0.25° for deriving soil parameters is shown in Fig. S2a. More detailed information on the VIC model calibration can be obtained from the previous studies (Mishra et al., 2010; Nijssen et al., 2001; Shah and Mishra, 2016a, 2016b).

### 2.2.2 The Noah Model

We used one-dimensional Noah model version 3.1(Mitchell, 2004; Schaake et al., 1996), which solves water and energy

balance in each grid cell. The Noah model has four soil layers. The model uses the modified Penman-Monteith equation to represent the diurnal variation of atmospheric resistance coefficient (Chen et al., 1996; Mahrt and Ek, 1984). In the Noah model, spatial variability of precipitation and infiltration is considered to estimate surface runoff, which is based on the exponential distribution of infiltration capacity. Baseflow is proportional to soil moisture storage. Vegetation parameters used in the Noah model were derived from the MODIS dataset and classified based on the Modified International Geosphere

Biosphere Programme (IGBP) scheme. The MODIS based IGBP product has 20 categories of land use/land cover data, which were derived during the observation period of 2001-2005. The vegetation parameters of the Noah model consist of vegetation fraction, stomatal resistance, minimum and maximum values of LAI, albedo, and roughness length. The major land cover classes are Forest, Shrubs, Savannas, Tundra, Grasslands, Croplands, Wetlands, Built-up, Ice, and Water. We used soil textures derived from digital soil map (Figure S2c) developed by Food and Agriculture Organization (FAO). The

forcing parameters required for the Noah model are daily precipitation, air temperatures (maximum and minimum), wind speed, surface pressure, relative humidity, surface downward long-wave radiation, and surface downward solar radiation. Daily meteorological forcing in the Noah model were internally disaggregated using uniform distribution for precipitation and diurnal cycle for other variables. We calibrated the Noah model parameters that include depth of four soil layers, Zilintikevich coefficient, surface runoff parameter, and bare soil evaporation component. Zilintikevich coefficient controls

the ratio of the roughness length for heat to the roughness length for the momentum, representing aerodynamic resistance.

### 2.2.3 The Community Land Model (CLM)

The CLM is a land surface component of community-developed global climate system model version 3.0 (CCSM v3.0), which was developed by the National Centre for Atmospheric Research (NCAR). CLM has 10 soil layers and similar to the VIC and Noah simulates both water and energy fluxes in each grid cell. Surface runoff in CLM is parameterized based on the TOPMODEL concept (Beven and Kirkby, 1979). Soil moisture storage in CLM is modelled after removing surface runoff, infiltration, and evaporation from surface storage. The basic difference in the CLM from the VIC and Noah is that the CLM has a representation of groundwater table, which is updated dynamically (Niu et al., 2007). The atmospheric forcing required for the CLM are daily precipitation, air temperatures (maximum and minimum), wind speed, specific humidity, incident solar radiation, and surface pressure. Land cover used in the CLM is represented by 17 plant functional types (PFTs), which were derived from MODIS and are classified using IGBP scheme similar to Bonan et al., 2002, while the soil textures (Fig. S2) used in CLM are derived from FAO datasets. We calibrated the soil thickness parameter for the CLM model similar to the VIC model. A detailed comparison of input parameters is provided in Table S1. All the three LSMs were run without considering irrigation and groundwater extraction as our aim was to understand the role of atmospheric forcing on root-zone (60 cm depth) soil moisture drought uncertainty.

### 2.3 Drought Indices

We used Standardized Precipitation Index (hereafter SPI; McKee et al., 1993) and Standardized Soil moisture Index (SSI; Hao and AghaKouchak, 2013) to represent meteorological and soil moisture (agriculture) droughts, respectively. We used 60 cm soil depth as a representative of root-zone soil moisture (Shah and Mishra, 2015). Since depths of root-zone and soil layers are different in all the three LSMs (Table S1), we estimated 60 cm soil moisture for each grid cell and for each LSMs, separately. A parametric (Gamma) distribution was fitted to precipitation and root-zone soil moisture to estimate SPI and SSI, respectively. For both SPI and SSI, the cumulative distribution functions obtained by fitting the Gamma distribution were mapped onto the normal distribution functions to represent a dimensionless index and derive drought indices (see Shah and Mishra (2015) and appendix A in supplemental material for more detail). We note that there are other approaches for estimating soil moisture drought index – for example a non-parametric percentile based drought index (Samaniego et al, 2013), but in this study we used a parametric (Gamma) distribution for estimating SSI so to be consistent with the precipitation based drought index (SPI).

### 2.4. Intensity - Areal extent - Frequency curves

Intensity-areal extent-frequency (IAF) curves for drought events were constructed to understand the frequency and severity of droughts in India for the period 1951-2015. The IAF curves were estimated using the root-zone soil moisture from the three LSMs (i.e. VIC, Noah, and CLM). We estimated drought severity using 4-month SSI at the end of the monsoon and

Rabi seasons (so as to represent the entire season) for the entire India and for the Indo-Gangetic plain region (longitude: 75-90ºE and latitude 23-30ºN; Figure S1) considering the climatological period of 1951-2015. The method to construct IAF curves has been described in detail in Mishra et al., (2016) and Mishra and Cherkauer (2010). IAF curves were estimated using the following steps: (i) for each year, mean 4-month SSI value was estimated for all the grids for areal extents of 2, 5, 10, 20, 30, 40, 50, 60, 70, 80, 90, and 100%, (ii) for each threshold of areal extent, mean severity of root-zone soil moisture drought was estimated for each year during the1951-2015 period, (iii) the Generalized Extreme Value (GEV) distribution was fitted to the mean severity for the selected areal extents and parameters (shape, location, and scale) were estimated using the maximum likelihood method, (iv) drought severity was estimated for the selected return periods of 2, 5, 10, 20, 25, 50, 100, 200 and 500 years for each areal extent threshold to construct IAF. Using IAF curves, mean intensity of drought can be estimated or for a given areal extent and frequency of drought. We evaluated the goodness of fit of the GEV distribution using QQ plots and Chi-Square goodness of fit test (supplemental Fig. S15-S17 and Table S11-S13).

## 3.0 Results

### 3.1 Calibration and evaluation of Land surface models (LSMs)

The three land-surface models (VIC, Noah, and CLM) parameterizations were manually constrained (calibrated) against observed streamflow across a set of eighteen major river basins covering approximately the entire landmass of India (see Figure S1; and Table S2). The performance of the LSMs in capturing the temporal dynamics of monthly streamflow during calibration and validation periods is quite satisfactory for most of the river basins (Table S2). The median correlation, $r$ (and Nash-Sutcliffe efficiency; NS) values estimated across these basins during the calibration period are around 0.91 (0.78), 0.90 (0.70), and 0.90 (0.70) for the VIC, Noah, and CLM, respectively. A similar level of (median) skill is also observed during the validation period (Table S2). The skill of the multi-model averaged streamflow of the three LSMs is comparatively better than that of individual models – with an overall median $r$ (and NS) value estimated across all basins is 0.91 (0.80) and 0.94 (0.77) during the calibration and validation period, respectively. The better performance of ensemble mean of simulated streamflow from the three LSMs against the observations shows the importance of considering ensemble mean of model outputs from the various land surface hydrological models. We also notice a relatively poor skill for all three LSMs and the ensemble mean in the coastal basins (e.g, Cauvery and East-coast basins – Table S2), which could be attributed to a number of factors including errors in forcing data and model parameterizations. Nevertheless, considering the wide range of hydro-climatic gradient across India, the efficiency of the three LSMs for capturing the observed streamflow can be considered reasonable.

Next we evaluated the skill of each model for capturing the observed dynamics of near surface and 60 cm soil moisture (Fig. 1; S3-S4). We used three different sources of soil moisture observations for this comparison purpose. The first set consisted of the weekly soil moisture observations taken at 18 IMD-based stations during the monsoon (JJAS) season for the period

2009-2013 (Unnikrishnan et al., 2013). The model simulated 60 cm soil moisture dynamics were compared against observations, which generally revealed a good skill for all three models (Fig. 1). Model simulated soil moisture showed a relatively higher correlation with observations in northern and western region as compared to those located in the southern coastal belt. Among models, the Noah simulated soil moisture exhibited higher correlation as compared to other two LSMs.

For this set-up, we also compared the calibrated vs un-calibrated model runs to understand what improvements (if any) could be achieved by the parameter calibration to simulation of soil moisture anomalies. We find limited benefits of the model calibration in this case – only the VIC model benefited by the model calibration mainly in the northern region locations and few of southern locations.

The second set of evaluation considered the continuous soil moisture observation datasets at an IIT Kanpur site available from the International Soil Moisture Network (ISMN: Dorigo et al. (2011)). Although all three models exhibited a general bias in capturing absolute values of the observed soil moisture, their daily variability observed over the course of the year is well captured by all three models (Fig. S5). Moreover the Noah and the CLM models show improvements in terms of reducing overall bias as a result of model calibration.

Finally, our third set of model evaluation considered an assessment of the model skill for capturing the remote sensing based soil moisture available from the ESA-CCI product (Dorigo et al, 2012). Here we used the modelled top-layer (10-30 cm) annual soil moisture over the period 1979-2012 for the comparison (Fig. S6). Despite the limitation that the ESA-CCI soil moisture inference is for the top few cm of earth surface, we find a positive correlation with modelled soil moisture for all
three models across a large part of India. A relatively higher correlation (more than 0.6) can be noticed for regions in the northwest and southern peninsular part of India.

We also evaluated the skills of LSMs for the terrestrial water storage (TWS) anomalies from the Gravity Recovery and Climate Experiment (GRACE – release v5.0; Landerer and Swenson 2012) derived products (1°×1°) for the period 2002-
2015. We used the ensemble mean of three available GRACE-TWS products (GeoForschungsZentrum, GFZ; Potsdam, Germany, Centre for Space Research at the University of Texas at Austin, USA, and Jet Propulsion Laboratory, USA) to reduce the noise (and scatter) among different TWS products. We compared ensemble mean GRACE-TWS against the monthly anomalies of modelled total column soil moisture from each of the three LSMs and their ensemble mean. The modelled total column soil moisture was aggregated to 1° spatial resolution to match the (coarse) resolution of the GRACE-
TWS product. Overall, all the three LSMs are able to capture the temporal dynamics of GRACE-TWS anomalies reasonably well across a large part of India (Fig. S7). The median correlation estimated across the modelled grid cells is more than 0.6 for all three LSMs – and the ensemble mean of simulated total column soil moisture anomalies showed an overall best (median) skill. All three LSMs (and the ensemble mean) exhibited a systematically lower performance in the northwest part of India (Fig. S7), which is most probably related to groundwater pumping effects that are not modelled in LSMs but are

captured in GRACE datasets (Asoka et al., 2017). Each LSM shows a slightly different area (grid cells) with the best skill score that motivates the use of multi-model ensemble mean to capture the (GRACE-based) water storage anomalies across a large part of India.

## 3.2 Multimodel ensemble droughts in India

We estimated areal extent of severe to exceptional droughts (SPI <-1.3) based on 4-month SPI at the end of the monsoon season (representing accumulated precipitation for June to September period) for the period of 1951-2015 (Fig. 2a). The top five monsoon season (JJAS) drought events occurred in 1987 (areal extent of severe to exceptional droughts: 35 %), 2002 (33.5 %), 1979 (27.7%), 1972 (26.3%), and 2009 (24.6%) at all India level. The monsoon season drought of 2015 (with an areal extent of 17.4%) and 2014 (14.4%) ranked 8[th] and 10[th] during the period of 1951-2015. Mishra et al. (2016) reported that the 2014-2015 monsoon season drought in the Indo-Gangetic plain was the most severe during the history of 116 years with a return period of 542 years.

We find that uncertainty in simulated areal extent of soil moisture drought (4-month SSI at the end of the monsoon season) estimated based on three LSMs is moderate during the monsoon season and has a year-to-year variability (Fig. 2a). For example, the one standard deviation representing the uncertainty in the simulated areal drought extent is on average estimated to be around 1.6%. However, during 1972 and 1979 monsoon seasons, uncertainty in areal extent of drought is 8% and 14 %, respectively (Fig. 2a; see also Supplementary Table S3). We estimated ensemble mean (ENS-SSI) areal extent of 60 cm soil moisture drought from the three LSMs for the monsoon season and found that 1987 (36.7%), 2002 (35.9%), 2009 (31.5%), 1972 (29.8%) and 1965 (23.4%) are the top five drought years during the period of 1951-2015 (Fig. 2a). We notice that 1979 (areal extent: 20%) ranked 6[th] on record while 2015 (16.5%) and 2014 (7.13%) ranked 8[th] and 15[th], respectively based on ENS-SSI areal extent of the simulated 60 cm soil moisture drought during the monsoon season.

Since the Rabi season (NDJF) is the key crop growing season in India, we estimated areal extents of 60 cm soil moisture drought from the three LSMs for the period of 1951-2015 (Fig. 2b). While year-to-year variability in the uncertainty of areal extent of droughts was found, the uncertainty in 60 cm soil moisture based the areal-extent of droughts was substantially higher (~ 5%) during the last decade of 1951-2015, which might be associated with frequent drought events during this period. We found a high uncertainty (~7%) in the areal extent of soil moisture drought in 2011 (Fig. 2b). Based on ENS-SSI areal extent of 60 cm soil moisture drought, the top five drought years occurred in 2003 (areal extent 30%, uncertainty: 9.6%), 2001 (27.6%, 12.0%), 1966 (22.6%, 4.7%), 1973 (20.7%, 8.9%), and 1988 (20.6%, 6.1%). Droughts in the Rabi season can be driven by both monsoon season precipitation deficit and positive temperature anomalies during this season. Uncertainty in the top five drought events in the Rabi season was substantial (5-12%), which underscores the need for multimodel drought assessment in the growing season.

The VIC, Noah, and CLM show all-India median autocorrelation of 0.23, 0.37, and 0.71, respectively at 4-month lag (Fig. 3) indicating that the CLM has the highest persistence in the 60 cm soil moisture. Spatial differences in soil moisture persistence were also observed (Fig. 3). Our results are in agreement with the findings of Wang et al. (2009) who reported

higher persistence for CLM modelled soil moisture, which can be attributed to its higher water holding capacity and thicker soil column (Fig. S2). Fig S2 shows that though there is not much differences in soil texture provided as input to three LSMs, available water in total column is much higher for the CLM, whereas We find that the spatial pattern of change in 60 cm soil moisture persistence after calibration matches quite well to with the spatial pattern of changes in soil layer thickness after calibration (Fig. S8). The same thing can be noted even considering the first three-soil layers which covers 60 cm soil

column for all three LSMs, the soil layer thickness of CLM dominates over other two LSMs. (Fig. S9).

Furthermore we find that all India averaged mean monthly soil moisture is the highest during July in the VIC and Noah LSMs, which is consistent with the seasonal cycle of all-India averaged precipitation (Fig. 3a-c). However, all India averaged mean monthly 60 cm soil moisture reaches to the highest level in the month of August in the CLM model (Fig. 3a-

c). The 1-month lag between peak precipitation and peak 60 cm soil moisture from the CLM can be due to a relatively deeper soil column (Fig. S8) and higher total column water holding capacity, more number of soil layers and difference in processes related to soil hydrology as discussed in Wang et al., (2009) and Xia et al. (2012).

To understand the relationship between meteorological and agricultural droughts, lagged correlation analysis was performed

between 4-month SPI at the end of the monsoon season and 4-month SSI (at the end of JJAS, JASO, ASON and so on). We find that 4-month SSI at the end of the monsoon season for the VIC model showed the highest correlation with 4-month SPI (JJAS) while 4-month SSI from the Noah model showed the lowest correlation (Fig. S10a). These results indicate that the 60 cm soil moisture from the VIC model responds faster to the monsoon season precipitation than the other two LSMs, which can be associated with soil moisture persistence and model parameterization (Van Loon et al., 2012; Wang et al., 2009; Xia

et al., 2012). However, we notice that the correlation between 4-month SPI at the end of the monsoon season and 4-month SSI declines rapidly after October (ONDJ, NDJF and so on) for the VIC and the Noah models (Fig. S10a). On the other hand, the CLM shows substantially higher persistence even for the March-June 60 cm soil moisture, which can be attributed to deeper soil column and differences in the other processes related to soil hydrology. These results also indicate that the anomalous precipitation during the monsoon season can last longer and have substantial influence on the agriculture drought

estimated using the 60 cm soil moisture from the CLM as reflected by the strength of the relationship between 12-month SPI and 12-month SSI in CLM (Fig. S10b).

Areal extent and severity of agricultural droughts estimated using the 60 cm soil moisture show a considerable uncertainty mainly due to the differences in soil moisture persistence characteristics among three LSMs (Fig. 3). We estimated areal

extent of agriculture drought from the three LSMs considering the period that showed maximum correlation against the monsoon season precipitation (Fig S10c). For instance, 4-month SSI at the end of October (JASO) showed the highest correlation with 4-month SPI at the end of September (JJAS) for the VIC and Noah models (Fig. S10a). On the other hand, 4-month SSI at the end of November (ASON) showed the highest correlation with the 4-month SPI at the end of September (JJAS) for the CLM. Therefore, we considered root-zone soil moisture for JASO, JASO, and ASON periods from the VIC, Noah, and CLM, respectively (Fig S10c) to understand the response of the monsoon season deficit in precipitation on agricultural drought. We find that the uncertainty in areal extent of agricultural drought is substantially reduced considering the lagged response of the monsoon season precipitation and soil moisture (Fig. S10c and Table S3) indicating that the major source of uncertainty in areal extent of agricultural droughts is soil moisture persistence in the LSMs.

### 3.3. Reconstruction of major droughts

We reconstructed major monsoon season drought events over India using 60 cm soil moisture from the three LSMs for the period of 1951-2015 (Fig 4). The meteorological and agricultural droughts were represented using 4-month SPI and 4-month SSI, respectively at the end of the monsoon season. We estimated ensemble mean 4-month SSI from the three LSMs to understand if the individual LSMs show a larger difference from the ensemble mean. Based on 4-month SPI at the end of the monsoon season, we selected the top two most widespread drought events that occurred in 1987 and 2002 (Fig. 2a and Fig. 4). Moreover, we also selected a recent drought event (2015) that caused an enormous water crisis in the Indo-Gangetic Plain (Mishra et al., 2016). For all the three major droughts (1987, 2002, and 2015) in the monsoon season, we compared areal extents of 60 cm soil moisture drought in the monsoon season estimated from the three LSMs.

Areal extents of the monsoon season droughts (meteorological and agricultural) in 1987, 2002, and 2015 show that droughts were mainly caused by the monsoon season precipitation deficits (Fig. 4). We notice positive air temperature anomalies in all the three years (1987, 2002, and 2015), however, patterns of agricultural and meteorological droughts were largely similar (Fig. 4). Among the three drought events, the monsoon season air temperature anomaly (positive) was the strongest in the 2015 monsoon season. Uncertainty in areal extent of agricultural droughts estimated using the 60 cm soil moisture is presented in supplemental Table S3. We notice large uncertainty in the areal extent of drought simulated from the three LSMs for the 2015 event (Table S3). The VIC model simulated areal extent of soil moisture drought was 14% during the 2015 monsoon season, while the areal extent of drought simulated from the Noah and CLM was 21.2 and 18.1%, respectively (Table S3).

Similar to the monsoon season droughts, we compared the spatial pattern of droughts simulated by the three LSMs for major droughts in the Rabi season, which occurred in1966, 1973, 2001, and 2003 (Fig. S11). We notice that major droughts in the Rabi season were also largely driven by the precipitation deficit and role of positive air temperature anomalies was relatively minor (Fig. S11). Overall, the VIC model shows the lesser intensity of drought during the Rabi season as compared to the

Noah and CLM for all the years, which can be attributed to differences in soil moisture persistence in the three LSMs (Fig. 3a). We find higher uncertainty in areal extent of drought during the 2001 Rabi season (Fig. S11) than other years of major drought events, which may be due to higher impacts of air temperature on drought during 2001. The overall uncertainty in the areal extent of agricultural droughts estimated using the 60 cm soil moisture estimated from the three models is presented in supplemental Table S4. In 2001, the areal extent of droughts simulated by the VIC, Noah, CLM, and their ensemble mean were 17.2, 40.7, 24.7, and 26.1%, respectively (Table S4). Large differences in the areal extent of 60 cm soil moisture drought simulated from the three LSMs were also noted for the year 1966, 1973, 2001, and 2003 (Table S4).

## 3.4 Intensity-Areal Extent-Frequency (IAF) of Droughts

Uncertainty in the multimodel drought estimates was evaluated using IAF curves (Fig. 5). Drought intensity associated with the selected areal extent thresholds were estimated for the return periods of 10, 20, 50, 100, 200, and 500 years (Fig. 5). We find that multimodel based drought intensity for a selected areal extent has a much larger uncertainty when the 95% confidence interval of the GEV parameters was considered (Table S6). However, uncertainty in drought intensity was lower when only mean values of the GEV parameters for drought intensity and areal extents was considered. Uncertainty in drought intensity appeared to grow with an increase in the return period (Fig. 5). Based on the IAF curves, we find that the 2002 monsoon season drought has a return period of 50 years. Moreover, uncertainty in drought intensity from the three LSMs was larger for smaller areal extents (Fig. 5). For instance, a drought of 50% areal extent and 50 year return period has intensities of -1.66, -1.91, and -1.53 simulated from the VIC, Noah, and CLM, respectively (Table S5). On the other hand, a drought of 5% areal extent and 50 year of return period has intensities of -2.89, -3.79, and -3.09 simulated from the VIC, Noah, and CLM, respectively (Table S5). Most of the return periods and areal extents drought intensities were higher for the Noah model and lower for the VIC model (Table S5), which can be associated with the differences in 60 cm soil moisture persistence in the three LSMs.

A considerably higher uncertainty in IAF curves during the Rabi season was noticed (Fig. S12 and Table S7-S8). For instance, a drought of 50% areal extent and 50 year return period can have intensities of -1.23, -1.62, and -1.50 for 60 cm soil moisture obtained from the VIC, Noah, and CLM, respectively (Fig S12 and Table S7). A drought of 5% areal extent and 50 year return period has intensities of -2.17, -3.71, and -3.25 simulated from the VIC, Noah, and CLM, respectively (Table S7). Higher uncertainty in drought intensities during the Rabi season can be attributed to the response of soil moisture in the three LSMs to meteorological forcing (precipitation and air temperature).

As the Indo-Gangetic plain is one of the most intensive crop growing regions in the world, we evaluated the uncertainty in the IAF curves constructed using the 60 cm soil moisture from the three LSMs (Fig. S13). We used the 12-month SSI at the end of December based on the areal averaged mean annual 60 cm soil moisture over the Indo-Gangetic plain. Similar to IAF

curves for the all-India averaged 60 cm soil moisture, a large uncertainty was found due to differences in the GEV parameters (Fig. S13, Table S9-S10). Moreover, uncertainty in IAF curves of the Indo-Gangetic plain increases with the return period and declines with an increase in areal extent of droughts. For instance, for an aerial extent of 5% and return period of 50 years, drought intensities were -2.59, -3.06, and -2.48 for the VIC, Noah, and CLM (Table S9). However, when

areal extent increases to 50%, drought intensities are -1.42, -1.49, and -1.45 for the VIC, Noah, and CLM, respectively (Table S9, Fig.S11). Overall, uncertainty in drought intensity is higher for localized droughts that have higher return periods. These results further indicate that 60 cm soil moisture drought characteristics can have large uncertainty arising from different LSMs, which can be reduced by considering the multimodel ensemble agricultural drought assessments.

**3.5 Role of the monsoon season precipitation and Rabi season air temperature**

We evaluated the differences in the coupling of 60 cm soil moisture (SSI) with monsoon season precipitation and air temperature (Fig. 6 and Fig. S14). We find that a 4-month SSI at the end of the monsoon season from the VIC model is strongly coupled (correlation coefficient =0.90) with the monsoon season precipitation over India. On the other hand, the 60 cm SSI showed correlation coefficients of 0.79 and 0.74 against the monsoon season precipitation for the Noah and CLM,

respectively (Fig. 6; c and e). These results show differences in the response of 60 cm SSI against changes in the monsoon season precipitation. However, all-India averaged 60 cm SSI showed stronger coupling with the monsoon season air temperature for the Noah and CLM models (correlation = -0.65 and -0.67) than that of the VIC model (correlation =-0.53) (Fig. 6; b, d, f). Interestingly, the coupling between 60 cm SSI for the Rabi and monsoon season precipitation is stronger for the CLM (correlation =0.76) and Noah (correlation =0.66) than that of the VIC model (correlation =0.55). These results

indicate that the monsoon season precipitation deficit can have a larger influence on the Rabi season drought in the CLM and Noah models compared to that of the VIC model. Similarly, the Rabi season air temperature showed a stronger relationship with 60 cm SSI for the CLM (correlation = -0.51) and Noah (-0.37) than that of the VIC model (-0.31). Similar differences in the Rabi and monsoon season 60 cm SSI for the Indo-Gangetic plain were observed for the three LSMs (Fig. S14), indicating that the drought indices based on the 60 cm soil moisture may show different sensitivity to atmospheric forcing, which can

show a substantial variation across regions and seasons.

**4. Discussion**

We find that the three LSMs show major differences in agricultural droughts during the monsoon and Rabi seasons. Uncertainty in the intensity of droughts is higher in the Rabi season than that of the monsoon season, which can be

associated with the role of air temperature on soil moisture. Also, we found differences in soil moisture drought simulated in response to precipitation and temperature deficits (Fig. 6 and S12). For instance, soil moisture from the VIC model shows a quick response to precipitation deficit whereas Noah and CLM show a delayed response. Moreover, localized droughts have

more uncertainty than that of wide-spread droughts over India and Indo-Gangetic Plain. The primary cause of the uncertainty in 60 cm soil moisture and droughts is related to soil moisture persistence (Fig. S16), which is associated with the soil water holding capacity as reported in Wang et al. (2009). We found that persistence in soil moisture is strongly linked with soil layer thickness (see Fig. S8), which in-turn affects the soil water holding capacity (available water, Fig. S16). Apart from the soil moisture persistence, there can be several other factors that can introduce uncertainty in 60 cm soil moisture simulations. For instance, all the three LSMs have different calibration parameters, which do not cover the entire range of uncertainty due to manual calibration (De Lannoy et al., 2006; Samaniego et al, 2013). However, our results show that the model calibration has a little impact on soil moisture anomalies which are largely driven by the climate forcing. Moreover, during drought, there is a high degree of non-linearity, therefore, calibration parameters estimated through global optimization may also not yield the best results (De Lannoy et al., 2006). The differences in vegetation parameters in the three LSMs can also attribute to uncertainty in 60 cm soil moisture (Peters-Lidard et al., 2008). For instance, the Noah model do not account for the sub-grid variability of vegetation, unlike the CLM and the VIC models, which use a mosaic based representation of vegetation. However, since we were interested about 60 cm soil moisture droughts, we assume that the major uncertainty in soil moisture simulations is due to soil hydraulic properties and the thickness of soil column in different LSMs (Peters-Lidard et al., 2008; Teuling et al., 2009). With respect to the LSMs, we would like to note that the drought uncertainty assessments conducted here are limited to only three LSMs, which is comparatively a smaller size.

Disparities in soil moisture persistence in the three LSMs can have implications for real-time drought monitoring and forecast. For instance, Shukla et al. (2013) reported that hydrologic initial conditions play a major role in hydrological prediction skills at a global scale. Similar findings were noted by Shah et al. (2017a) who found that hydrological initial conditions play a vital role in prediction skills of soil moisture droughts over India. Hydrologic prediction at short to seasonal scales can be influenced by soil moisture persistence and the LSMs with higher persistence can have more skill contributed by the initial hydrologic conditions. This further highlights a need of multimodel based real-time drought monitoring and prediction system over India as shown in Wang et al. (2009). We found that multimodel ensemble mean performs better than individual LSMs for streamflow and terrestrial water storage (TWS) from GRACE. Bohn et al. (2010) reported that multimodel ensemble average may not always yield a higher prediction skill at seasonal scales, however, at shorter lead times, it can provide better confidence in prediction of soil moisture droughts due to higher skill from hydrologic initial conditions (Shukla et al. 2013; Shah et al. 2017a). We also find disparities in coupling between monsoon/Rabi season 60 cm soil moisture and precipitation/air temperature. The differences in soil moisture sensitivity to atmospheric forcings can have implications for future projections of droughts from multiple LSMs. For instance, Prudhomme et al. (2014) reported that uncertainty in drought projections can be large especially due to models that simulate the dynamic response of plants to climate. Overall, we find that 60 cm soil moisture droughts have uncertainty associated with their areal-extent and severity. The uncertainty in drought estimates is largely due to differences in the soil moisture persistence. Uncertainty in

drought estimates during the crop-growing season can be reduced using the multimodel ensemble mean, which can assist decision makers in India.

## 5. Conclusions

India has witnessed some of the most severe meteorological and agricultural droughts during the period of 1951-2015. The most wide-spread meteorological droughts during the monsoon season occurred in 1987, 2002, 1972, 1979, and 2009. During the Rabi season, the most wide-spread agricultural droughts occurred in 2003, 2001, 1966, 1973, and 1988. All the three LSMs, as well as their ensemble mean, identified major 60 cm soil moisture droughts between 1951 and 2015.

All the three LSMs (e.g VIC, Noah, and CLM) showed differences in persistence of 60 cm soil moisture over India, which was largely associated with soil water holding capacity. The CLM showed the highest soil moisture persistence among the three LSMs. Due to differences in the soil moisture persistence, areal extent and intensity of droughts from the three LSMs showed uncertainty. Using the IAF curves, we found that the uncertainty in intensity was higher for the localized droughts
(with less areal extent). Uncertainty increases with the return period of droughts indicating that localized and rare drought events have more differences in the three LSMs.

All the three LSMs showed differences in the coupling between 60 cm soil moisture and precipitation/air temperature suggesting that LSMs have disparities in soil moisture sensitivity to precipitation and temperature anomalies in the monsoon
and Rabi seasons. Considering the differences in drought characteristics simulated from the three models, multi-model ensemble mean can be a better estimate of agricultural droughts over India as demonstrated for streamflow and terrestrial water storage. Uncertainty in intensity and areal extent can be reduced substantially for the severe and localized droughts that can affect agricultural production. Future studies should consider soil moisture simulations from more number of LSMs as well as other sources of uncertainty in the historical reconstruction of agricultural droughts over India. Moreover,
including the uncertainty due to choice of different precipitation datasets and other meteorological forcing datasets can be important for regional drought impacts assessments.

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

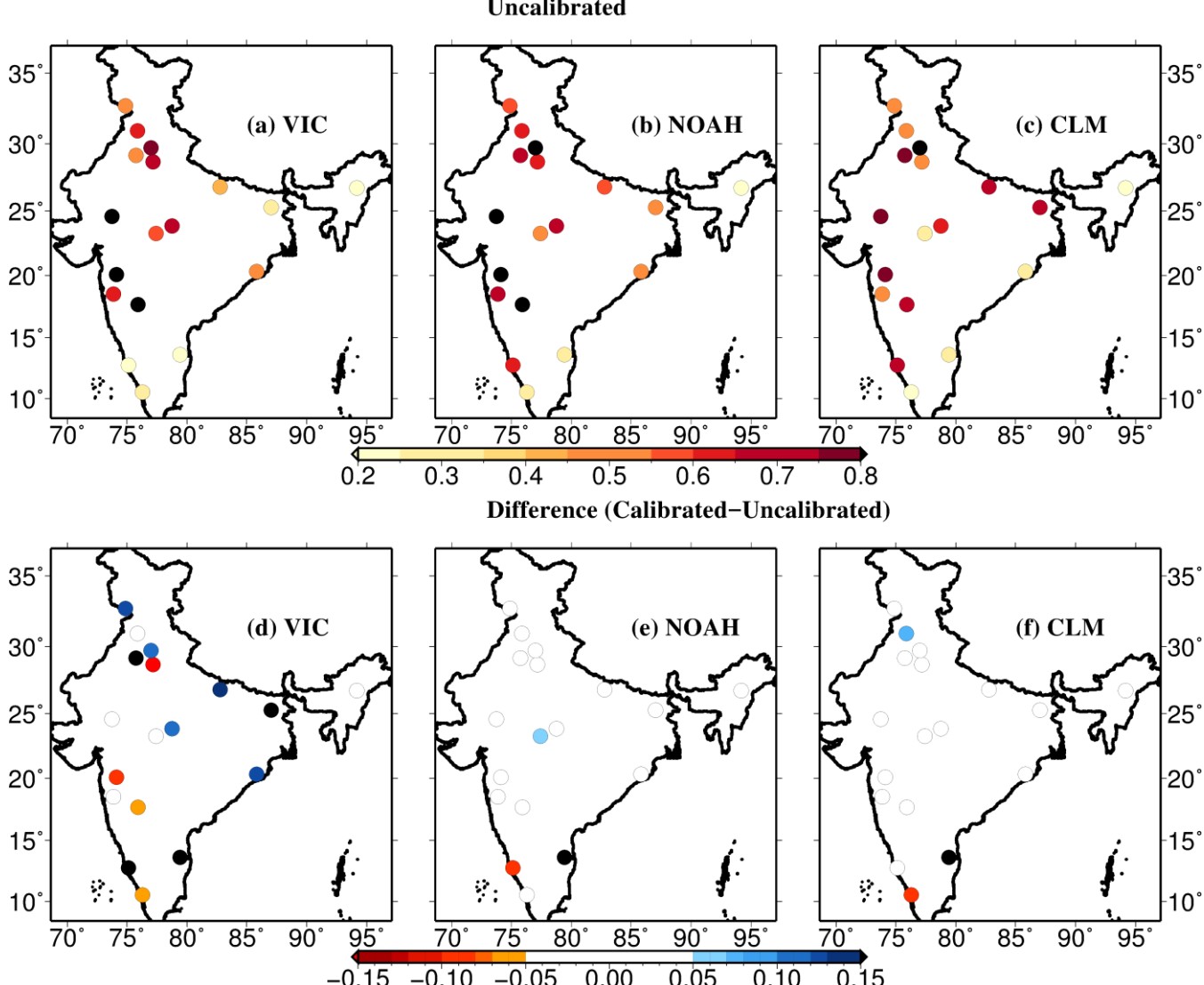

Figure 1. Correlation of weekly 60 cm simulated soil moisture with IMD gauge based soil moisture (~60 cm) during the monsoon season, 2009-2013. (a-c) Correlation for control (default or uncalibrated) set-up for the VIC, NOAH, and CLM, respectively. (d-f) Show difference (calibrated-uncalibrated) in correlation coefficient.

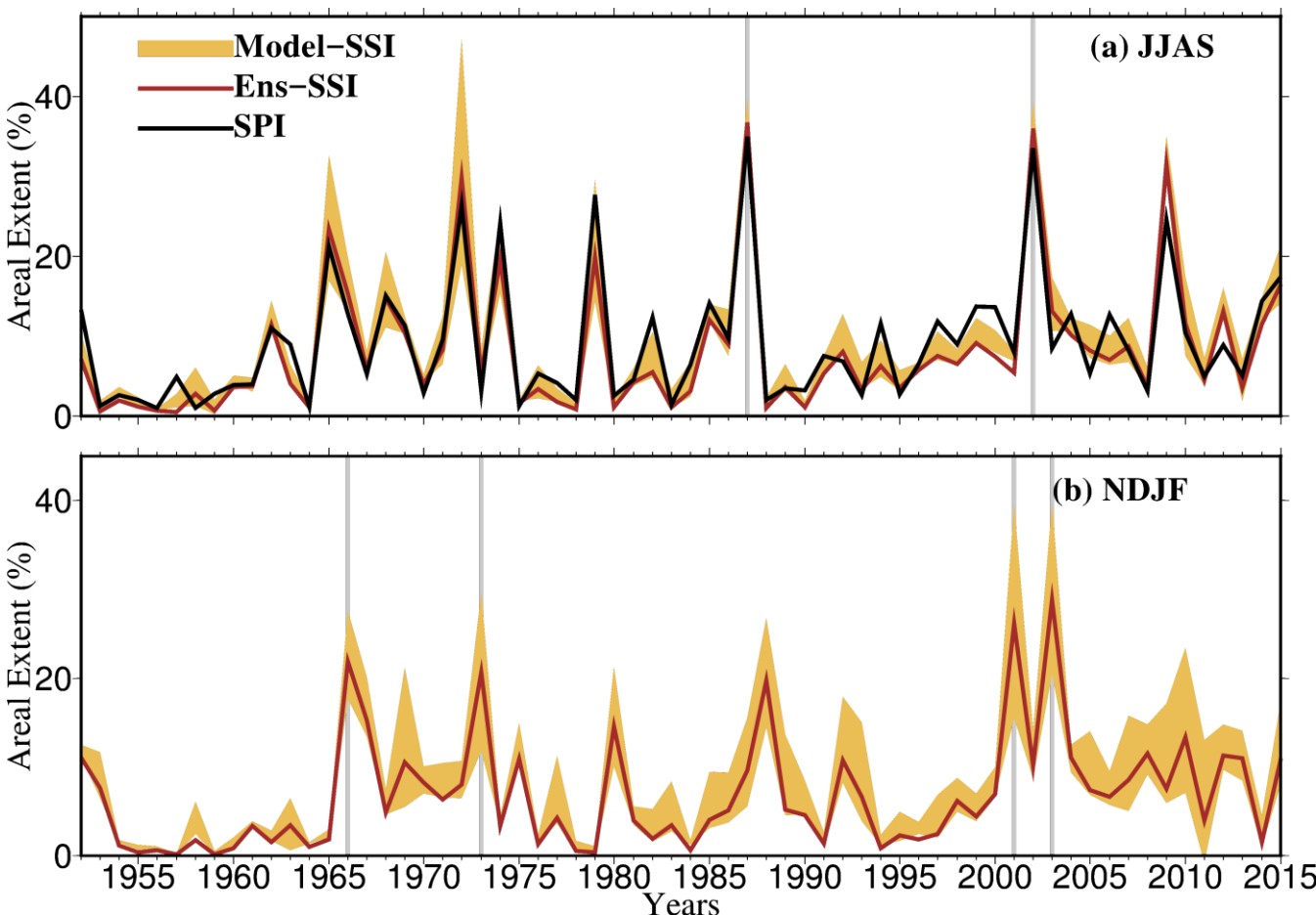

Figure 2: Uncertainty in areal extent (%) of 60 cm soil moisture drought simulated using the three LSMs (i.e. VIC, Noah, and CLM). (a) Multimodel ensemble (brown) mean 4-month Standardized Soil Moisture Index (SSI) and inter-model variation (shaded) estimated as one standard deviation for the monsoon season. Black line in (a) shows 4-month Standardized Precipitation Index (SPI) at the end of the monsoon season (June through September) (b) multimodel ensemble mean and uncertainty in 4-month SSI estimated using the three LSMs for the Rabi season (November through February). Light brown shaded area shows uncertainty in severe-to-exceptional drought based on model simulated SSI (SSI <-1.3). Dark brown line shows areal extent estimated based on ensemble mean SSI for the three LSMs. Grey line marks top drought years based on area under drought.

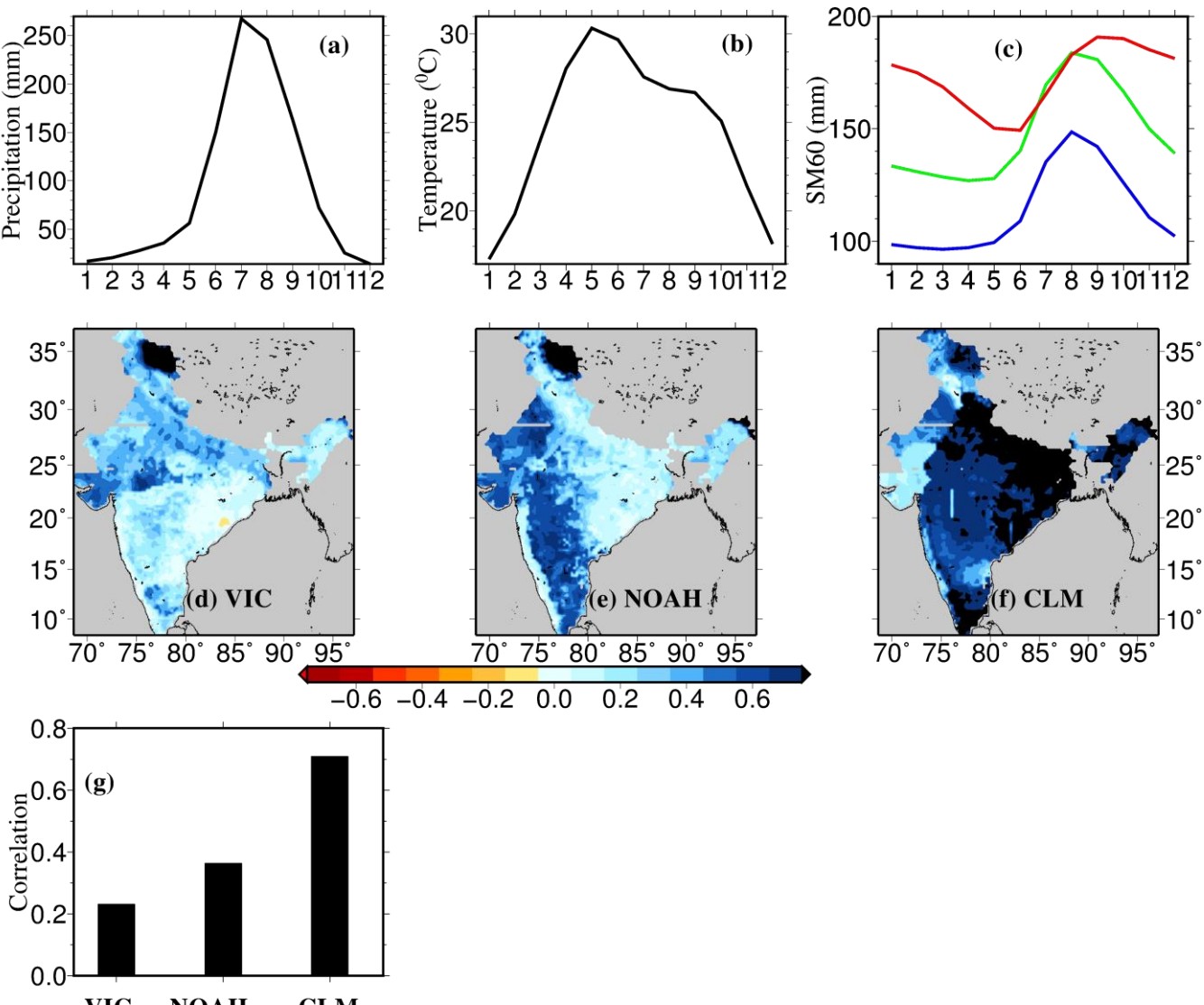

Figure 3: Uncertainty in persistence in root-zone soil moisture (60 cm). Seasonal cycle of all-India averaged (a) precipitation (b) mean air temperature and (c) 60 cm soil moisture simulated using the VIC (blue), the Noah (green), and the CLM (red). (d,e,f ) Autocorrelation in 60 cm soil moisture at 4-month lag simulated using the VIC, Noah, and CLM, respectively. (g) All-India median autocorrelation (4-month lag) in the 60 cm soil moisture from the three LSMs.

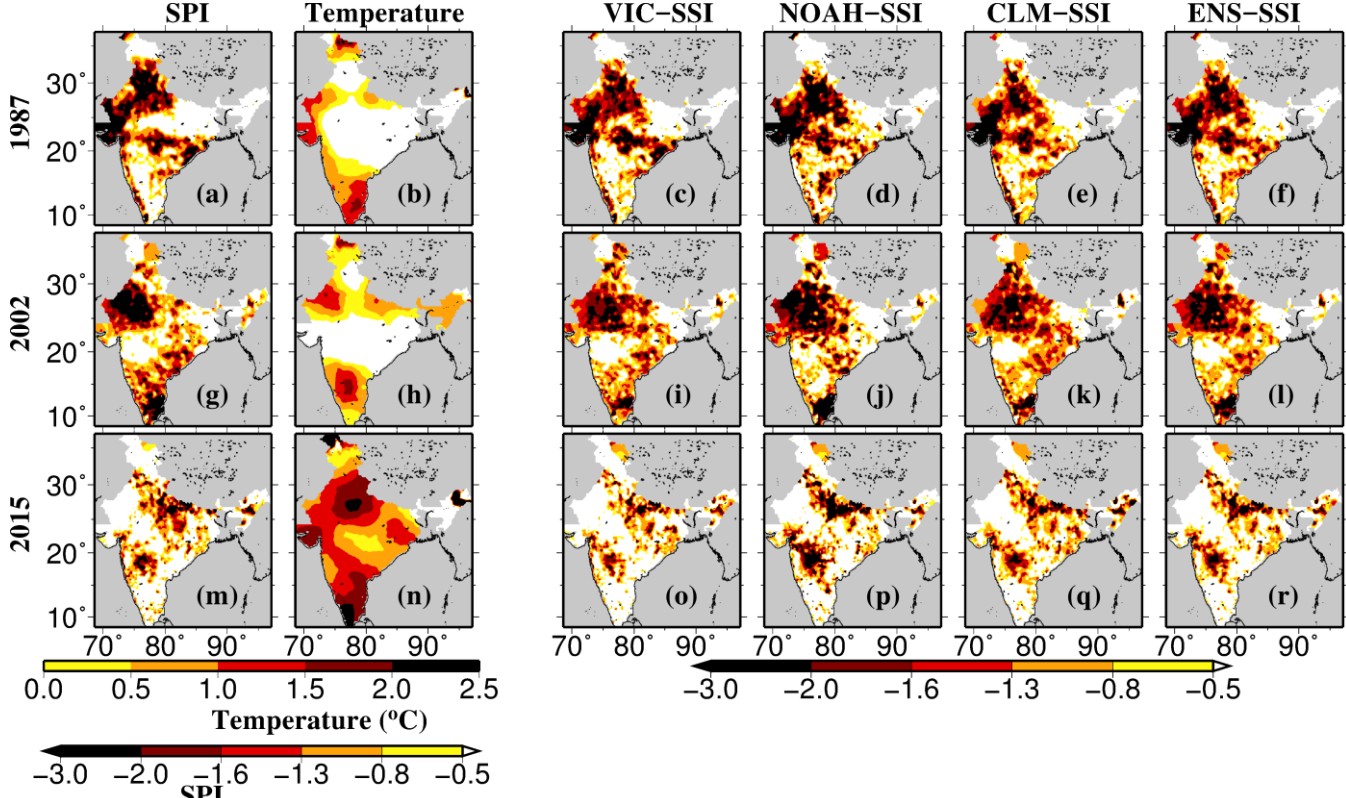

Figure 4: Reconstruction of monsoon season drought events of (a-f) 1987, (g-l) 2002 and (m-r) 2015, estimated based on (a,g,k) 4-month SPI at the end of the monsoon season, (c,i,o) 4-month SSI at the end of the monsoon season simulated using the VIC model, (d,j,p) same as (c,i,o) but for the Noah model, and (e,k,q) same as (c,i,o) but for the CLM. (f,l,r) Ensemble mean 4-month SSI simulated using the VIC, Noah, and CLM. (b,h,n) Air temperature anomaly during the monsoon season for the selected drought years.

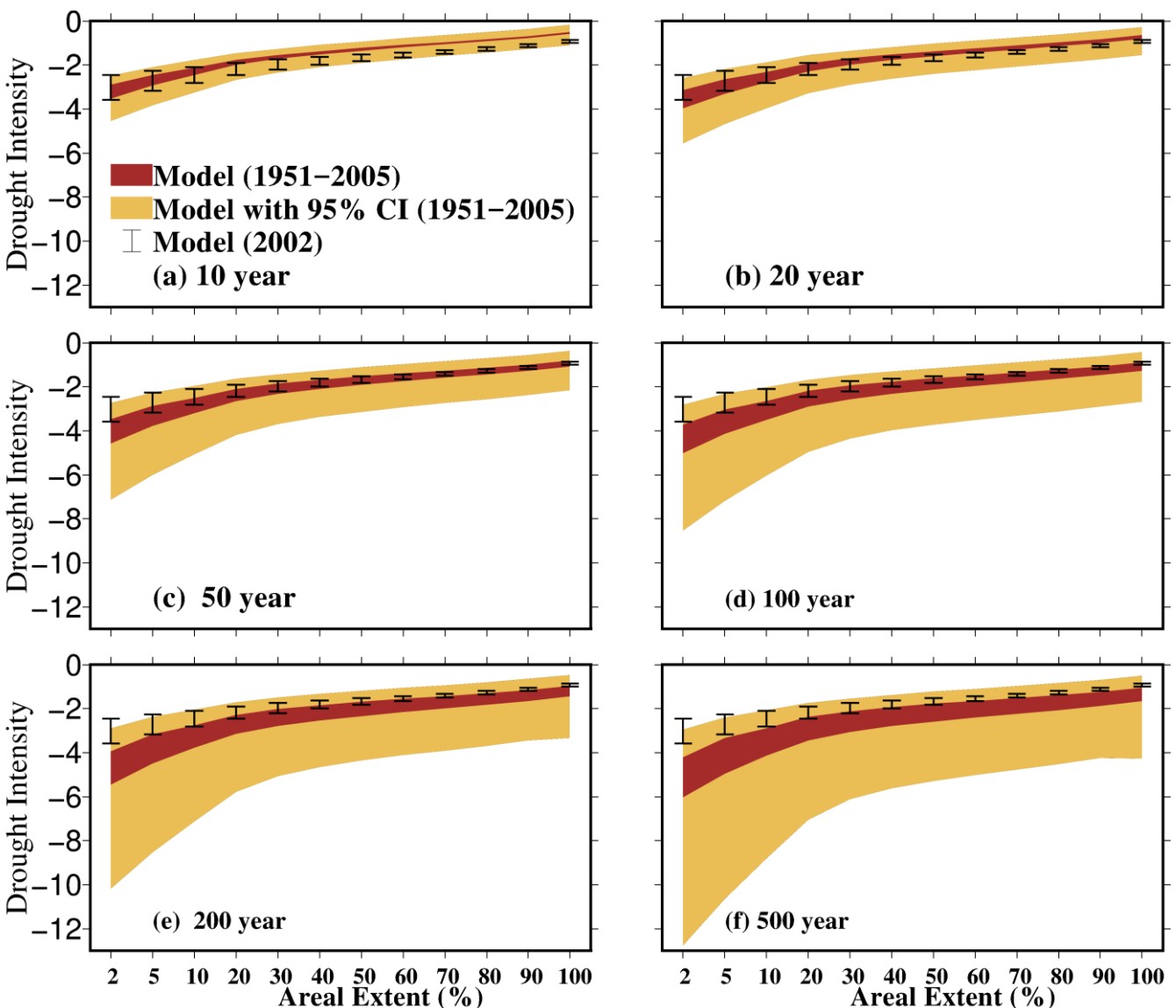

Figure 5: Uncertainty in Intensity-Areal Extent-Frequency (IAF) curves for the monsoon season 60 cm soil moisture drought estimated using the three LSMs. Dark brown color shade shows uncertainty in models without considering parameter uncertainty in the Generalized Extreme Value (GEV) distribution while light brown color shows uncertainty considering 95% confidence interval of the GEV parameters for return periods (a) 10, (b) 20, (c) 50, (d) 100, (e) 200, and (f) 500 years. Black error-bars indicate uncertainty for the 2002 monsoon season drought using three LSMs.

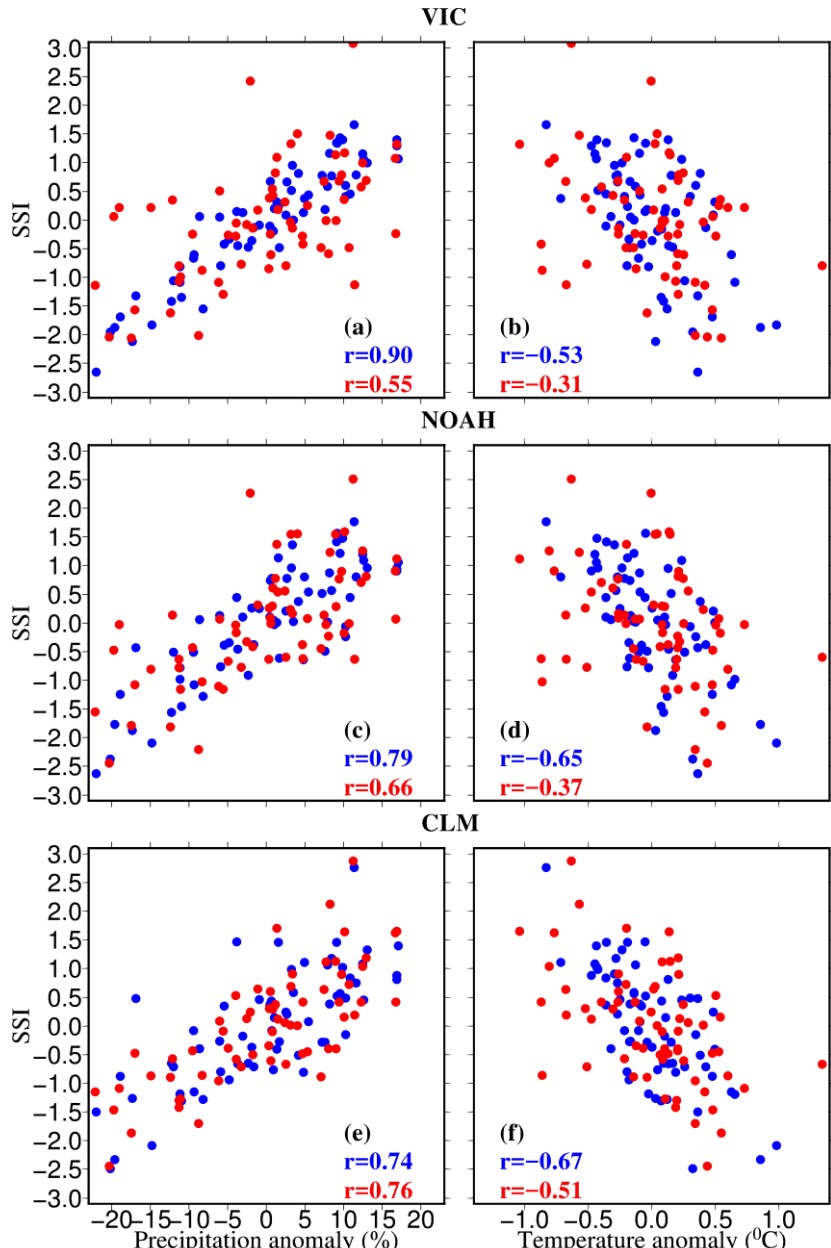

Figure 6. (a,c,e) Relationship between monsoon season precipitation anomaly (%) and 4-month SSI at the end of the monsoon season; and (b,d,f) same as (a,c,e) but for the relationship between 4-month SSI and air temperature anomaly of the monsoon season. Correlation coefficients are shown for all-India SSI (blue) and 4-month SSI over the Indo-Gangatic Plain (red).