# Peer review of "Reconstruction of droughts in India using multiple land surface models (1951-2015)"

_Hydrology and Earth System Sciences, 2017_

## Referee Comment (RC1) · Anonymous Referee #1 · 29 Jul 2017

Review of the paper "Reconstruction of droughts in India using multiple land surface models (1951-2015)" by Mishra et al. (hess-2017-302) submitted to Hydrol. Earth Syst. Sci.

The authors manually calibrate three land surface models (CLM, Noah, VIC) by using observed monthly streamflow at streamflow gauges for one period ($\sim$ 5 years) and validate for the independent other period ($\sim$ 5 years). The three models were run from 1951 to 2015 to produce root zone ($\sim$60 cm) soil moisture products. The authors use these soil moisture products to analyze Indian agricultural drought events including severity, frequency, and drought extent. The results found that there is larger uncertainty in crop growing season than the monsoon season. The large uncertainty is mainly due to the difference in model parameterizations – different soil moisture

persistence. The results suggest using multi-model ensemble for Indian drought monitoring. For model setup and calibration, model evaluation, and analysis of differences in model parameterizations, the paper shows some major deficiencies in its general appearance. Therefore, I recommend a major revision of the manuscript.

Major comments:

1. Model setup

(a) Spin-up period: Is a spin-up period run? If no, please explain reason. If so, how long is run for each model for this spin-up period? Was soil moisture equilibrium state including deep soil layer checked? (b) I do not think that you can use daily meteorological forcing data to run Noah and CLM? In general, hourly surface forcing data are used to drive such land surface models. How to divide daily meteorological forcing data into hourly time scale? (c) It is not clear how to calibrate Noah model. Why are depth of soil layers, Zilintikevich coefficient, surface runoff parameter and bare soil evaporation component selected? Is any sensitivity test performed or does the selection just depend on your own experience? Which are surface runoff parameter and soil evaporation component? What possible values do you use? How to manually tweak these values for each basin individually or together? I am puzzling how to calibrate soil layer depth. Based on my experience, the Noah four soil layers are 0-10 cm, 10-40cm, 40-100 cm, and 100-200 cm. The mid-layer is 5 cm, 25 cm, 70 cm, and 150 cm. If you calibrate soil depth, for each grid point at a given basin, you adjust these soil layer depths. If so, can you make a plot to compare these calibrated soil depths with default soil layer depths. (d) It is very confused how to calibrate CLM using soil depth layers. More explanations are needed. (e) What are soil parameters in Section 2.2.2 and 2.2.3? Are they soil textures (types)? Noah and CLM use the soil textures derived from FAO, and VIC uses soil texture derived Harmonized World Soil Moisture Datbase (HWSD). I am wondering how big differences exist between two datasets? It is very well known that different texture has different soil related parameters such as field capacity, wilting point, etc., which leads to different temporal variation. (f) Different

vegetation type classification datasets are used for different models, which can result in additional uncertainty for soil moisture product as different vegetation type has different root zone (leads to different transpiration even though surface meteorological forcing is the same). (g) There is only one test in this study – calibrated run. I would like to see the control run/default run (the default parameters are used) and the comparison with the calibrated run. This will demonstrate what benefits you gain from the calibration process.

2. Model evaluation

(a) Calibrated model is only evaluated against observed streamflow. Unfortunately, I am very disappointed that the soil moisture used in this study is not evaluated against either in-situ observations or remotely sensed soil moisture. There are a few stations in India to measure soil moisture from different datasets such as Global Soil Moisture Data Bank (Robock et al. 2000), In-situ observations of soil moisture from India Meteorological Department (Unnikrishnan et al. 2016), and international soil moisture network (https://ismn.geo.tuwien.ac.at/). In addition, quite a few of remotely sensed soil moisture products such as SMAP, SMOS, SMOPS, ASCAT, AMSR2 and more are not used to evaluate LSMs soil moisture simulation products. However, the major variable used in this study is 60 cm soil moisture. Robock, A., et al., 2000: The Global Soil Moisture Data Bank, BAMS, 81, 1281-1299. Unnikrishnan, C. K., et al., 2016: Validation of two gridded soil moisture products over India with in-situ observations, J. Earth System Science, 125, 935-944. (b) The authors assumed 60 cm soil layer as root zone. However, for each individual model, it defines its root zone varying from vegetation type to vegetation type. For example in Noah, grass root zone is 1m and forest is 2m. I suggest the authors use 60 cm soil moisture in whole text to avoid confusing the readers.

3. Model result analysis

(a) The uncertainty analysis is very limited due to three models as the samples are

too few for a representative of model uncertainties. In general, the spread can roughly show an uncertainty range when three-model ensemble is used. The authors need to indicate this weakness in a discussion section. (b) The authors indicated that the uncertainty in soil moisture is mainly due to model parameterizations – resulting in different persistence of soil moisture. They assumed that there is a large field capacity for CLM but there is no further investigation. In practical, different soil texture datasets, different vegetation type classification datasets, different model structure (specific soil layer in CLM and Noah vs hydrological soil layer concept), and other ET parameterizations may affect this uncertainty together. I recommend make several sensitivity tests to clarify these issues. At least, plot field capacity, wilting point, soil type, vegetation type, root zone depth for all models and then compare their differences. (c) In Figure 3c, the seasonal cycles in Noah and VIC are comparable although the magnitude is quite different. However, that in CLM is completely different with Noah and VIC. This further suggests that soil moisture evaluation against in situ observations and remotely sensed product is needed to identify which is closer to the observations. (d) In line 33, page 7, the authors cited Wang et al. (2009) to explain higher persistence in soil moisture due to larger water holding capacity and thicker soil column. However, the authors used 60 cm soil layer for all models and also need plot water holding capacity for top 60 cm to verify this point. (e) The authors find an interesting point, that is, there are larger uncertainties in Rabi season than monsoon season. Unfortunately, the authors do not make further investigation to look for the reason. They use a general sentence "which can be associated with the role of air temperature and precipitation on soil moisture" to explain. When Figure 4 and Figure S3 are checked, during the monsoon season, three models have larger similarity than Rabi season mainly due to VIC model. A possible reason is that VIC water mode rather than energy mode is used in this study. During the monsoon season, water is unlimited and limited energy is used due to less net radiation (rainy and cloud sky). Energy and water-mode type model does not have big difference. However, during Rabi season, water is limited but energy may be unlimited, so that energy-type model (Noah, CLM) shows larger difference than water-mode type

model (VIC). A quick check is to use VIC energy mode to re-run this test to compare with VIC water mode run.

Minor Comments:

1. Check Table S1: Surface downward shortwave and longwave radiation , for CLM v3.0, soil texture based on IGBP or FAO or vegetation type data based on IGBP. 2. Check Table S2: East coast, calibration and validation period is overlapped. 3. Check Table S2: Mahanadi, calibration and validation period is overlapped. 4. Check Table S2: Subarmarekha, calibration and validation period is overlapped.

I assumed that the authors used independent period to validate the calibrated models. If not, please explain the reason.

---

## Referee Comment (RC2) · Anonymous Referee #2 · 18 Aug 2017

In this study, three models were implemented to conduct watershed simulation in India. The amazing thing is that the whole India was included, however, quite a few modeling details were missing. Therefore, I probably cannot proceed detailed review at this point. I would suggest adding those details as supplementary information in the next round. On the other hand, there are other similar work done by using multiple models (not limited: Scavia et al. (2017)I Sharifi et al. (2017). You did not mention the advantages/disadvantages by using multiple models (and, why these three models???). It cannot always only for the good reasons right? Overall, the content of the given manuscript is way less than it should be (in all sections). Good luck in the next round.

- Scavia, D., M. Kalcic, R. L. Muenich, J. Read, N. Aloysius, I. Bertani, C. Boles, R. Confessor, J. DePinto, M. Gildow, J. Martin, T. Redder, S. Sowa, Y. Wang, H. Yen, 2017.

[Figure]

Multiple SWAT models guide strategies for agricultural nutrient reductions. Frontiers in Ecology and the Environment, 15(3), pp. 126-132. - Sharifi, A., H. Yen, K. M. B Boomer, L. Kalin, X. Li, D. E. Weller, 2017. Using multiple watershed models to assess the water quality impacts of alternate land development scenarios for a small community. Catena, 150C, pp. 87-99.

---

## Referee Comment (RC3) · Anonymous Referee #3 · 28 Aug 2017

General comments: This manuscript demonstrate the method to reconstruct meteorological and soil moisture droughts in India by three LSMs, i.e. VIC, Noah, and CLM. The overall scientific idea is clearly expressed in detail. The manuscript could be considered to be published after the following minor concerns are addressed. And the language should be carefully polished and make the whole manuscript concise and precise. Specific comments: 1. Page 1 Line 8. In Abstract, "As a large population of India is dependent on agriculture, soil moisture droughts adversely affect agriculture and groundwater resources " This sentence is illogical and should be rephrased. 2. Page 5 Line 5. The definition and the formula of SPI and SSI should be clearly expressed in the manuscript instead of just giving the cited literature and leaving the readers to the literature. In another word, the manuscript should be self-contained. 3. Page 5

[Figure]

Line 18. It would be better if the Indo-Gangetic Plain Region can be shown in supplemental figure. 4. Page 6 Line 20. How do you compare the TWS (which is the total terrestrial water storage including both surface water, soil moisture and groundwater) and the total column soil moisture (which is just the soil moisture stored in aquifers)? 5. Page 7 Line 5. Why the ranks of 2014 and 2015 droughts identified in this manuscript are different from the cited literature Mishra et al., 2016b. Which study is confirmed by the in-situ records and which is biased identification? This affects the quality of the drought reconstruction in this manuscript and should be addressed carefully. 6. Page 8 Line 2-5. What do you mean by "soil depths were calibrated in all the three LSMs"? Are the soil depths in LSMs the same or not? If they are the same, "The 1-month lag between peak precipitation and peak root-zone soil moisture from the CLM can be due to a relatively deeper soil column." Should be deleted. If not, "since......" should be deleted. Above all, you should clearly express the reason and not confuse with each other: soil depths, the number of soil layers or processes related to soil hydrology. Technical corrections: 1. Page 1 Line 25. The citation Mishra et al., 2016b comes first before 2016a. The literature should be cited in order of their appearance in manuscript. 2. Page 3 Line 4. Digital elevation map (DEM) should be rewritten as digital elevation model (DEM).

---

## Referee Comment (RC4) · Anonymous Referee #4 · 31 Aug 2017

Review for "Reconstruction of droughts in India using multiple land surface models (1951-2015)" by Mishra et al. in Hydrology and Earth System Sciences

RECOMMENDATION A major revision

SUMMARY: This study reconstructed past droughts over India using multiple land surface models (LSMs). Standardized Precipitation Index (SPI) and Standardized Soil moisture Index (SSI) were used for detection and characterization of meteorological and agricultural drought, respectively. In this study, root-zone soil moisture was estimated from VIC, Noah, and CLM. The parameters of each LSM were calibrated. This study found that there are larger uncertainties in agricultural droughts over a large part of India during crop growing seasons than during monsoon seasons. This study concluded that different persistence of soil moisture from the three LSMs are caused by

the difference in model parameterization. Overall, the manuscript is written well but some words and sentences are necessarily revised due to misuses and grammatical errors. The topic is a good-fit to Hydrology and Earth System Sciences (HESS), but I have several major comments on the method and findings. Also, there are several minor comments on the scientific representations, especially figures. More details of the major comments are listed below. Due to the major issues, the current version of the manuscript is not publishable in the HESS. Therefore, I recommend major revision.

General Major Comments:

It has been very popular to compare the estimated hydro-climate variables from different climate or land surface models (e.g. CMIP3 and CMIP5). One of the lessons from the previous inter-comparison studies is that it is hard to understand what really happens in the models (more likely a black box) unless common parameters (e.g., infiltration capacity or vegetation fraction) across the models and their impacts on the interest estimate (herein, root zone soil moisture (down to 60 cm) are evaluated. In this study, there is a missing section for evaluations of simulated soil moisture, before converting soil moisture to SSI, which give valuable information for how different soil moisture dynamics are across the models. Also, there is a missing for comparisons of the common parameters, which can bring a fundamental understanding of the sensitivity of root-zone soil moisture to the common parameters even though this study discussed that soil water holding capacity (a common parameter) plays an important role in soil moisture dynamics. Therefore, adding sections for root-zone soil moisture analysis and parameter comparison is strongly recommended.

Uncertainties from a different combination of parameter sets even within one model can bring certain uncertainties in drought estimates, which can question the relative importance of calibration methods or physics (representation of important processes in a model) on soil moisture estimation.

In addition, the output from three LSMs are not able to provide a full distribution of the

root-zone soil moisture estimates due to different model structures and parameters. The method introduced in this study might be appropriate for a sensitivity test of the simulated root-zone soil moisture to different land surface model structures and parameters. In Figure 2, the spreads of areal extents from three models were represented as the envelope but they are actual three points in each year. Or, the authors need to clarify the definition of uncertainty.

Minor comments:

Abstract: Page 1 Line 13: "higher uncertainty" should be replaced with "higher sensitivity."

Page 1 Line 18: "multi-model ensemble" should be replaced with "multi-model average." The ensemble is often used for different perturbed physics, initial condition, and forcing within one model.

Page 1 Line 23: "severity" should be replaced with "intensity" for consistency with the later section.

Page 2 Line 29-30: What are the temporal coverage of precipitation from 6995 gage stations from IMD? Have the IMD precipitation products compared with the CRU and GPCC (even though they are 0.5 degree)? It is worth to understand how large the uncertainties in precipitation from different sources are.

Page 4 Line 15-18: Zilintikevich coefficient and its explanation should be placed at the end of the sentence.

Page 5 Line 10-11: Is a Gamma (parametric) distribution appropriate in computing a agricultural (soil moisture) drought index? What about using percentiles (nonparametric) as a drought index?

Page 5 Line 17: Why this study uses the 4-month SSI? I assume that it was matched with Rubi seasons but there is no explanation about it. Please clarify it.

Page 5 Line 20: Drought severity is defined as the total area (intensity x duration) from initiation through recovery. "for each year, mean severity of droughts" is confusing. Please change the sentence as "for each year, mean 4-month SSI value was . . ."

Page 6 Line 7: "the ensemble mean streamflow" should be replaced with "the multi-model averaged streamflow."

Page 7 Line 11-12: What is the definition of uncertainty in area extent of drought? Please clarify it either here or in the method section.

Page 8 Line 32-33: How can higher persistence of CLM soil moisture can be attributed to its higher water holding capacity and thicker soil column? Please explain the possible physical processes. The explanation will be beneficial for readers.

Page 9 Line 21-22: This study finds that regardless of seasons, precipitation is a major driver for drought and temperature is a minor. If then, uncertainties in meteorological forcings, especially precipitation might be more important than uncertainties in soil moisture. Why didn't this study investigate uncertainties in precipitation?

Page 11 Line 1-15: Please discuss the potential implementation of the findings in the section 3.5.
* * *

---

## Author Comment (AC1) · 7 Sep 2017

**Response to Reviewer #1 comments**

The authors manually calibrate three land surface models (CLM, Noah, VIC) by using observed monthly streamflow at streamflow gauges for one period (_~5 years) and validate for the independent other period (~5 years). The three models were run from 1951 to 2015 to produce root zone (~60 cm) soil moisture products. The authors use these soil moisture products to analyze Indian agricultural drought events including severity, frequency, and drought extent. The results found that there is larger uncertainty in crop growing season than the monsoon season. The large uncertainty is mainly due to the difference in model parameterizations – different soil moisture persistence. The results suggest using multi-model ensemble for Indian drought monitoring. For model setup and calibration, model evaluation, and analysis of differences in model parameterizations, the paper shows some major deficiencies in its general appearance. Therefore, I recommend a major revision of the manuscript.

We thank the reviewer for his/her insightful comments. We have made every possible effort to address the reviewer's comments in an adequate manner (in below).

1. Model setup:

(a) Spin-up period: Is a spin-up period run? If no, please explain reason. If so, how long is run for each model for this spin-up period? Was soil moisture equilibrium state including deep soil layer checked?

Yes, we run a spin-up period for all models to avoid (undesirable) influence of initial conditions. The spin-up period was set to 1951-2015. For this period, we ran each model and generated an initial state file using which the simulations were conducted for the entire period. Moreover, we performed an exploratory analysis to make sure that each model is in the stable condition (equilibrium state) from the beginning simulations.

(b) I do not think that you can use daily meteorological forcing data to run Noah and CLM? In general, hourly surface forcing data are used to drive such land surface models. How to divide daily meteorological forcing data into hourly time scale?

The variables other than daily precipitation, maximum and minimum temperatures, and wind speed were generated using the VIC model, which uses the MTCLIM method. As we have reported in our manuscript, the effectiveness of the MTCLIM algorithm has been evaluated using the observations from the flux-tower for various ecosystems of the world (Bohn et al. 2013). The Noah and CLM models internally disaggregate daily forcing to sub-daily time steps. The output from these models was again aggregated to daily time step for the analysis.
We would like to mention here that the method chosen for the disaggregation of daily values to hourly ones would not have any (substantial) effects on the resulting monthly soil moisture simulations, which is eventually used for the drought analysis.

(c) It is not clear how to calibrate Noah model. Why are depth of soil layers, Zilintikevich coefficient, surface runoff parameter and bare soil evaporation component selected? Is any sensitivity test performed or does the selection just depend on your own experience? Which are surface runoff parameter and soil evaporation component? What possible values do you use? How to manually tweak these values for each basin individually or together? I am puzzling how to calibrate soil layer depth. Based on my experience, the Noah four soil layers are 0-10 cm, 10-40cm, 40-100 cm, and 100-200 cm. The mid-layer is 5 cm, 25 cm, 70 cm, and 150 cm. If you calibrate soil depth, for each grid point at a given basin, you adjust these soil layer depths. If so, can you make a plot to compare these calibrated soil depths with default soil layer depths.

An initial sensitivity analysis was performed using one parameter at a time to identify the parameters that are sensitive to streamflow. After this analysis, we selected the parameters for calibration. Based on the prior studies (Hogue et al. 2005), we selected model parameters for the calibration. The selected parameters are tweaked manually (generating sets of model parameters and selecting the best among them based on model skill to represent observed stream flow). This is done individually for every selected river basin (see Supplement Fig. S1). We adjusted the soil depth following the (calibration) approach of the VIC model in which the soil depths are estimated via a calibration procedure such that the modeled stream-flow matches the observed values. We followed this approach given the wide success of the VIC model application in a wide variety of river basins across different climatic conditions. Moreover using a similar (calibration) approach, we aim to harmonize the different model applications over India. So, the soil depths in LSMs are treated as calibration parameters, which vary across the models and the river basins. We will make this issue clear in the revised manuscript.

As suggested, we will provide skill scores of every model using the calibrated parameters and default soil depths, which could further help illustrating the advantages of the model calibration.

(d) It is very confused how to calibrate CLM using soil depth layers. More explanations are needed.

Please refer to the above response. However, in the revised manuscript, we will discuss the calibration process in detail and the selection of the parameters.

(e) What are soil parameters in Section 2.2.2 and 2.2.3? Are they soil textures (types)? Noah and CLM use the soil textures derived from FAO, and VIC uses soil texture derived Harmonized World Soil Moisture Datbase (HWSD). I am wondering how big differences exist between two datasets? It is very well known that different texture has different soil related parameters such as field capacity, wilting point, etc., which leads to different temporal variation.

We appreciate the insightful comment from the reviewer. In the revised manuscript, we evaluated the difference in soil moisture simulations from the

VIC model using the soil parameters from the FAO and HWSD. We do not find any significant difference in these two set of parameters. The underlying (soil-textural) dataset for the HWSD is mostly derived based on the FAO datasets. Notably the different models use different pedo-transfer functions to derive soil related parameters and in this case even if the two models uses a same underlying dataset the resulting soil parameters can be different.

In the revised manuscript, we will show plots depicting similarity/differences among (common) soil parameters across LSMs.

(f) Different vegetation type classification datasets are used for different models, which can result in additional uncertainty for soil moisture product as different vegetation type has different root zone (leads to different transpiration even though surface meteorological forcing is the same).

Thanks. As stated above due to the different requirements of different models in terms of different soil and vegetation parameters, we are enforced to use different vegetation type classification datasets.

We will note this issue of additional sources of uncertainty due to requirement of different soil and vegetation datasets in the revised manuscript. However, we have evaluated the sensitivity of vegetation parameters derived from the different sources on drought assessment and results suggest no major difference.

(g) There is only one test in this study – calibrated run. I would like to see the control run/default run (the default parameters are used) and the comparison with the calibrated run. This will demonstrate what benefits you gain from the calibration process.

This is a good suggestion. For the drought assessment, the model calibration may not contribute significantly as we use only anomalies of soil moisture, which can largely be driven by the climate variations. As mentioned above, we will provide skill scores of every model using the calibrated and default parameters in the revised manuscript.

2. Model evaluation

(a) Calibrated model is only evaluated against observed streamflow. Unfortunately, I am very disappointed that the soil moisture used in this study is not evaluated against either in-situ observations or remotely sensed soil moisture. There are a few stations in India to measure soil moisture from different datasets such as Global Soil Moisture Data Bank (Robock et al. 2000), In-situ observations of soil moisture from India Meteorological Department (Unnikrishnan et al. 2016), and international soil moisture network (https://ismn.geo.tuwien.ac.at/). In addition, quite a few of remotely sensed soil moisture products such as SMAP, SMOS, SMOPS, ASCAT, AMSR2 and more are not used to evaluate LSMs soil moisture simulation products. However, the

major variable used in this study is 60 cm soil moisture. Robock, A., et al., 2000: The Global Soil Moisture Data Bank, BAMS, 81, 1281-1299. Unnikrishnan, C. K., et al., 2016: Validation of two gridded soil moisture products over India with in-situ observations, J. Earth System Science, 125, 935-944.

In the revised manuscript, we will show the model skill in representing observed dynamics of soil moisture (using both the observation dataset from Global Soil Moisture Data Bank and a proxy remote sensing derived gridded soil products). We however like to note here that most of the gridded (remotely sensed) soil moisture products have their inherent uncertainty – a quite prominent one is that they are limited in their inference of soil water to only few cm from ground surface).

(b) The authors assumed 60 cm soil layer as root zone. However, for each individual model, it defines its root zone varying from vegetation type to vegetation type. For example in Noah, grass root zone is 1m and forest is 2m. I suggest the authors use 60 cm soil moisture in whole text to avoid confusing the readers.

Thank you. We will revise the text following your suggestion.

3. Model result analysis

(a) The uncertainty analysis is very limited due to three models as the samples are too few for a representative of model uncertainties. In general, the spread can roughly show an uncertainty range when three-model ensemble is used. The authors need to indicate this weakness in a discussion section.

We will include this as a limitation in the revised manuscript.

(b) The authors indicated that the uncertainty in soil moisture is mainly due to model parameterizations – resulting in different persistence of soil moisture. They assumed that there is a large field capacity for CLM but there is no further investigation. In practical, different soil texture datasets, different vegetation type classification datasets, different model structure (specific soil layer in CLM and Noah vs hydrological soil layer concept), and other ET parameterizations may affect this uncertainty together. I recommend make several sensitivity tests to clarify these issues. At least, plot field capacity, wilting point, soil type, vegetation type, root zone depth for all models and then compare their differences.

We will include plots showing differences/similarity among (common) parameters across LSMs.

(c) In Figure 3c, the seasonal cycles in Noah and VIC are comparable although the magnitude is quite different. However, that in CLM is completely different with Noah and VIC. This further suggests that soil moisture evaluation against in situ

observations and remotely sensed product is needed to identify which is closer to the observations.

As mentioned above, we will include an evaluation plot for different LSMs in the revised manuscript.

(d) In line 33, page 7, the authors cited Wang et al. (2009) to explain higher persistence in soil moisture due to larger water holding capacity and thicker soil column. However, the authors used 60 cm soil layer for all models and also need plot water holding capacity for top 60 cm to verify this point.

We will include the plot of water holding capacity across different LSMs in the revised manuscript.

(e) The authors find an interesting point, that is, there are larger uncertainties in Rabi season than monsoon season. Unfortunately, the authors do not make further investigation to look for the reason. They use a general sentence "which can be associated with the role of air temperature and precipitation on soil moisture" to explain. When Figure 4 and Figure S3 are checked, during the monsoon season, three models have larger similarity than Rabi season mainly due to VIC model. A possible reason is that VIC water mode rather than energy mode is used in this study. During the monsoon season, water is unlimited and limited energy is used due to less net radiation (rainy and cloud sky). Energy and water-mode type model does not have big difference. However, during Rabi season, water is limited but energy may be unlimited, so that energy-type model (Noah, CLM) shows larger difference than water-mode type model (VIC). A quick check is to use VIC energy mode to re-run this test to compare with VIC water mode run.

Thank you for this comment. We will include this in the discussion of the revised manuscript as per your suggestion.

Minor Comments:

1. Check Table S1: Surface downward shortwave and longwave radiation , for CLM v3.0, soil texture based on IGBP or FAO or vegetation type data based on IGBP.

We will edit this in the revised manuscript.

2. Check Table S2: East coast, calibration and validation period is overlapped.

We will check this in the revised manuscript.

3. Check Table S2: Mahanadi, calibration and validation period is overlapped.

We will check this in the revised manuscript.

4. Check Table S2: Subarmarekha, calibration and validation period is overlapped.

We will check this in the revised manuscript.

I assumed that the authors used independent period to validate the calibrated models. If not, please explain the reason.

Yes we have used the independent period for the model calibration and validation. We will make this point clear in the revised manuscript.

---

## Author Comment (AC2) · 7 Sep 2017

**Response to Reviewer #2 comments**

In this study, three models were implemented to conduct watershed simulation in India. The amazing thing is that the whole India was included, however, quite a few modeling details were missing. Therefore, I probably cannot proceed detailed review at this point. I would suggest adding those details as supplementary information in the next round. On the other hand, there are other similar work done by using multiple models (not limited: Scavia et al. (2017) Sharifi et al. (2017). You did not mention the advantages/disadvantages by using multiple models (and, why these three models???). It cannot always only for the good reasons right? Overall, the content of the given manuscript is way less than it should be (in all sections). Good luck in the next round.

- Scavia, D., M. Kalcic, R. L. Muenich, J. Read, N. Aloysius, I. Bertani, C. Boles, R. Confessor, J. DePinto, M. Gildow, J. Martin, T. Redder, S. Sowa, Y.Wang, H. Yen, 2017. Multiple SWAT models guide strategies for agricultural nutrient reductions. Frontiers in Ecology and the Environment, 15(3), pp. 126-132.

- Sharifi, A., H. Yen, K. M. B Boomer, L. Kalin, X. Li, D. E.Weller, 2017. Using multiple watershed models to assess the water quality impacts of alternate land development scenarios for a small community. Catena, 150C, pp. 87-99.

We thank the reviewer for his/her insightful comments. We will take care of the suggested references in the revised manuscript. By using the multiple models, we aim to represent the (model) uncertainty in soil-moisture drought simulations across India. We will enhance the discussion part of the revised manuscript to make it clear about advantages and limitations of using the multiple models. Moreover, we provide more details on the land surface models in the supplemental section of the revised manuscript.

---

## Author Comment (AC3) · 7 Sep 2017

**Response to Reviewer #3 comments**

This manuscript demonstrate the method to reconstruct meteorological and soil moisture droughts in India by three LSMs, i.e. VIC, Noah, and CLM. The overall scientific idea is clearly expressed in detail. The manuscript could be considered to be published after the following minor concerns are addressed. And the language should be carefully polished and make the whole manuscript concise and precise.

We thank the reviewer for his/her insightful comments. We have addressed the reviewer's comments and we will further check the manuscript for language and conciseness.

Specific comments:
1. Page 1 Line 8. In Abstract, "As a large population of India is dependent on agriculture, soil moisture droughts adversely affect agriculture and groundwater resources " This sentence is illogical and should be rephrased.

Thank you. We will reformulate this sentence in the revised manuscript.

2. Page 5 Line 5. The definition and the formula of SPI and SSI should be clearly expressed in the manuscript instead of just giving the cited literature and leaving the readers to the literature. In another word, the manuscript should be self-contained.

We will explicitly mention the definition (and formula) for SPI and SSI in the revised manuscript – as an Appendix or in Supplement.

3. Page 5 Line 18. It would be better if the Indo-Gangetic Plain Region can be shown in supplemental figure.

Thank you. We will show the Indo-Gangetic Plain region in the supplements.

4. Page 6 Line 20. How do you compare the TWS (which is the total terrestrial water storage including both surface water, soil moisture and groundwater) and the total column soil moisture (which is just the soil moisture stored in aquifers)?

We used the anomaly of the entire soil-water column for every model and compared it against the GRACE derived TWS anomaly. In doing so, we agree with the reviewer that we might have missed the groundwater component; and this might be relevant for a certain part of India. On the other hand, considering that the temporal dynamics of monthly groundwater is rather very slow, the most of the temporal variability in the TWS anomaly can be expressed by the soil-water part. Therefore, comparing the anomaly of the modeled soil water column with GRACE derived TWS anomaly for assessing the skill in terms of capturing the temporal variability, as considered in this study, would be reasonable. Here, our aim is not to predict TWS using soil moisture, instead we

just considered TWS as surrogate of observations to evaluate the models' skill for soil moisture simulation.

5. Page 7 Line 5. Why the ranks of 2014 and 2015 droughts identified in this manuscript are different from the cited literature Mishra et al., 2016b. Which study is confirmed by the in-situ records and which is biased identification? This affects the quality of the drought reconstruction in this manuscript and should be addressed carefully.

Here we think there is some misunderstanding. In the cited paper Mishra et al, 2016b; we looked for the muti-year drought ranking, rather than a single year drought (as done in this study). Moreover, here the ranking is based on soil moisture drought rather than meteorological drought as in Mishra et al. (2016). Soil moisture droughts are affected by the soil moisture persistence, therefore, ranking (from soil moisture and precipitation) may be different.

6. Page 8 Line 2-5. What do you mean by "soil depths were calibrated in all the three LSMs"? Are the soil depths in LSMs the same or not? If they are the same, "The 1-month lag between peak precipitation and peak root-zone soil moisture from the CLM can be due to a relatively deeper soil column." Should be deleted. If not, "since: : :: : :" should be deleted. Above all, you should clearly express the reason and not confuse with each other: soil depths, the number of soil layers or processes related to soil hydrology.

We followed the (calibration) approach of the VIC model in which the soil depths are estimated via a calibration procedure such that the modeled stream-flow matches the observed values. We followed this approach given the wide success of the VIC model application in a wide variety of river basins across different climatic conditions. Moreover using a similar (calibration) approach, we aim to harmonize the different model applications over India. So, the soil depths in LSMs are treated as calibration parameters, which vary across the models and the river basins. We will make this clear in the revised manuscript.

Technical corrections:

1. Page 1 Line 25. The citation Mishra et al., 2016b comes first before 2016a. The literature should be cited in order of their appearance in manuscript.

Thank you. We will edit this during the revision.

2. Page 3 Line 4. Digital elevation map (DEM) should be rewritten as digital elevation model (DEM).

Thank you. We will edit this during the revision.

---

## Author Comment (AC4) · 7 Sep 2017

**Response to Reviewer #4 comments**

This study reconstructed past droughts over India using multiple land surface models (LSMs). Standardized Precipitation Index (SPI) and Standardized Soil moisture Index (SSI) were used for detection and characterization of meteorological and agricultural drought, respectively. In this study, root-zone soil moisture was estimated from VIC, Noah, and CLM. The parameters of each LSM were calibrated. This study found that there are larger uncertainties in agricultural droughts over a large part of India during crop growing seasons than during monsoon seasons. This study concluded that different persistence of soil moisture from the three LSMs are caused by the difference in model parameterization. Overall, the manuscript is written well but some words and sentences are necessarily revised due to misuses and grammatical errors. The topic is a good-fit to Hydrology and Earth System Sciences (HESS), but I have several major comments on the method and findings. Also, there are several minor comments on the scientific representations, especially figures. More details of the major comments are listed below. Due to the major issues, the current version of the manuscript is not publishable in the HESS. Therefore, I recommend major revision.

We thank the reviewer for his/her insightful comments. We have addressed the reviewer's comments in the revised manuscript.

General Major Comments:

It has been very popular to compare the estimated hydro-climate variables from different climate or land surface models (e.g. CMIP3 and CMIP5). One of the lessons from the previous inter-comparison studies is that it is hard to understand what really happens in the models (more likely a black box) unless common parameters (e.g., infiltration capacity or vegetation fraction) across the models and their impacts on the interest estimate (herein, root zone soil moisture (down to 60 cm) are evaluated. In this study, there is a missing section for evaluations of simulated soil moisture, before converting soil moisture to SSI, which give valuable information for how different soil moisture dynamics are across the models. Also, there is a missing for comparisons of the common parameters, which can bring a fundamental understanding of the sensitivity of root-zone soil moisture to the common parameters even though this study discussed that soil water holding capacity (a common parameter) plays an important role in soil moisture dynamics. Therefore, adding sections for root-zone soil moisture analysis and parameter comparison is strongly recommended.

Thank you. We will include a section on "Soil moisture analysis and parameter comparison" in the revised manuscript. Here we will check the correspondence of different model simulated root-zone soil moisture along with characteristics like persistence and seasonal behaviors. Apart from this, we will evaluate the modeled soil moisture estimates against some proxy (satellite-based) soil moisture that could further shed some lights on individual modeled soil moisture

simulations. We will also include an analysis depicting the similarity/differences among the common model parameters related to soil moisture simulations – this could be root-zone soil water holding capacity, across India.

In addition, the output from three LSMs are not able to provide a full distribution of the root-zone soil moisture estimates due to different model structures and parameters. The method introduced in this study might be appropriate for a sensitivity test of the simulated root-zone soil moisture to different land surface model structures and parameters. In Figure 2, the spreads of areal extents from three models were represented as the envelope but they are actual three points in each year. Or, the authors need to clarify the definition of uncertainty.

We agree with the reviewer on the aspect that three chosen LSMs may not cover the full uncertainty of the root-zone soil moisture estimates. There are number of factors that need to be considered in understanding the full distribution of root-zone soil moisture estimates that include among other things, the uncertainty in forcing variables (precipitation, temperature, etc), land-surface variables (soil textural information and soil hydraulic parameters like porosity, field capacity and permanent witling point), as well as, model conceptual parameters.

Nevertheless with the application of three models, our aim in this paper was to show the differences in soil-moisture simulations and resulting drought characteristics over India. We term these differences to soil moisture simulations uncertainty to convey the main message that the application of a single model to study soil moisture droughts over India may not be adequate.

Furthermore, we will provide a note in the concluding paragraph of the revised manuscript on future efforts on including other LSMs or hydrological models for analyzing soil moisture drought analysis; as well as for conducting analysis to recognize the contribution from other sources of uncertainty.

Minor comments:
Abstract: Page 1 Line 13: "higher uncertainty" should be replaced with "higher sensitivity."

Thank you. We will revise the text as suggested.

Page 1 Line 18: "multi-model ensemble" should be replaced with "multi-model average." The ensemble is often used for different perturbed physics, initial condition, and forcing within one model.

Thank you. We will incorporate your suggestion in the revised manuscript.

Page 1 Line 23: "severity" should be replaced with "intensity" for consistency with the later section.

Thank you. We will revise it as suggested.

Page 2 Line 29-30: What are the temporal coverage of precipitation from 6995 gage stations from IMD? Have the IMD precipitation products compared with the CRU and GPCC (even though they are 0.5 degree)? It is worth to understand how large the uncertainties in precipitation from different sources are.

A detail description of the methodology and underlying dataset used to create the gridded IMD field is provided Pai et al (2014). In this study, we restricted to use the IMD dataset, which uses much more underlying station dataset than used in CRU or GPCC. As said earlier, our goal here is to understand the uncertainty in soil moisture due to usage of different LSMs; we restrict ourselves to use a single set of forcing dataset for all three LSMs.

As a side note, in our recent study, we have compared the IMD based precipitation product with the CRU ones for drought analysis (see Mishra et al., 2016b). On a long-time period the two products agree on showing similar basic features, but they do differ (sometime substantially) on a short-time period (see a recent paper by Jing and Wang (2017) Nature CC).

Page 4 Line 15-18: Zilintikevich coefficient and its explanation should be placed at the end of the sentence.

We will add a sentence on the coefficient as suggested.

Page 5 Line 10-11: Is a Gamma (parametric) distribution appropriate in computing a agricultural (soil moisture) drought index? What about using percentiles (nonparametric) as a drought index?

We have tested the appropriateness of the distribution function. We used here the parametric (Gamma) form for the agricultural drought so to be consistent with the precipitation based drought index (SPI). We, however, agree with the reviewer of using a more robust non-parametric based drought index (like percentile) – we adopted for such approach in the recent past (see Mishra et al, 2016b) – but here in this study we do not find big differences in our modeling results due to different approach of estimating drought indices (the main results and conclusions however remain unaffected).

Page 5 Line 17: Why this study uses the 4-month SSI? I assume that it was matched with Rubi seasons but there is no explanation about it. Please clarify it.

Yes, the reviewer is right in his/her interpretation. We will add a sentence to clarify this in the revised manuscript.

---

## Author Response (AR1)

We thank all the reviewers and the editor for their constructive and insightful comments that have helped in improving the revised version.

**Response to Reviewer #1 comments**

The authors manually calibrate three land surface models (CLM, Noah, VIC) by using observed monthly streamflow at streamflow gauges for one period (~5 years) and validate for the independent other period (~5 years). The three models were run from 1951 to 2015 to produce root zone (~60 cm) soil moisture products. The authors use these soil moisture products to analyze Indian agricultural drought events including severity, frequency, and drought extent. The results found that there is larger uncertainty in crop growing season than the monsoon season. The large uncertainty is mainly due to the difference in model parameterizations – different soil moisture persistence. The results suggest using multi-model ensemble for Indian drought monitoring. For model setup and calibration, model evaluation, and analysis of differences in model parameterizations, the paper shows some major deficiencies in its general appearance. Therefore, I recommend a major revision of the manuscript.

We thank the reviewer for his/her insightful comments. We have made every possible effort to address the reviewer's comments in an adequate manner (in below).

1. Model setup:

(a) Spin-up period: Is a spin-up period run? If no, please explain reason. If so, how long is run for each model for this spin-up period? Was soil moisture equilibrium state including deep soil layer checked?

Yes, we ran a spin-up period for all the land surface models to avoid (undesirable) influence of initial conditions. The spin-up period was set to 1951-2015. For this period, we ran each model and generated an initial state file using which the simulations were conducted for the entire period. Moreover, we performed an exploratory analysis to make sure that each model is in the stable condition (equilibrium state) from the beginning simulations. In this respect, we have included a sentence in the revised manuscript as "*All the three LSMs were first spun-up using data of 65 years (1951-2015) to establish initial conditions for the modelled states and fluxes.*" on page 3 lines 32-33.

(b) I do not think that you can use daily meteorological forcing data to run Noah and CLM? In general, hourly surface forcing data are used to drive such land surface models. How to divide daily meteorological forcing data into hourly time scale?

All three LSMs were forced with daily meteorological forcings. We have mentioned following in revised manuscript "*All the theree LSMs have sub-routines to disaggregate daily precipitation uniformly to sub-daily time scale, while temperature and radiation are temporally disaggregated following the diurnal cycle.*" on page 3, lines 21-23. Finally the outputs from these models were again aggregated to monthly time-steps for further analysis. More information on disaggregation process of the meteorological forcing can be obtained from

http://www.hydro.washington.edu/Lettenmaier/Models/VIC/Documentation/p_disag.shtml

http://www.hydro.washington.edu/Lettenmaier/Models/VIC/Documentation/VICDisagg.shtml

(c) It is not clear how to calibrate Noah model. Why are depth of soil layers, Zilintikevich coefficient, surface runoff parameter and bare soil evaporation component selected? Is any sensitivity test performed or does the selection just depend on your own experience? Which are surface runoff parameter and soil evaporation component? What possible values do you use? How to manually tweak these values for each basin individually or together? I am puzzling how to calibrate soil layer depth. Based on my experience, the Noah four soil layers are 0-10 cm, 10-40cm, 40-100 cm, and 100-200 cm. The mid-layer is 5 cm, 25 cm, 70 cm, and 150cm. If you calibrate soil depth, for each grid point at a given basin, you adjust the soil layer depths. If so, can you make a plot to compare these calibrated soil depths with default soil layer depths.

Based on the prior studies (Hogue et al. 2005) initially we identified a set of model parameters for calibrating the Noah model. We also identified soil depth as calibration parameter following success of similar technique used to calibrate the VIC model (Nijssen et al. 2001). Then, a first-order sensitivity analysis was performed using one parameter at a time to identify the parameters that are sensitive to streamflow. After this analysis, we selected final parameters for calibration. The selected parameters are tweaked manually (generating sets of model parameters and selecting the best among them based on model skill to represent observed streamflow). This is done individually for every selected river basin (see Supplement Fig. S1). Here we would like to make a note that for the Noah model, soil-depth for the Indus basin was only calibrated as to achieve a reasonable model skill. We show the inferred soil-layer thickness for first three soil layers after calibration in Fig. 1 (below); and the change in the total soil-column depth after calibration (compared to default) in Fig. 2 (below).

In this respect, we have included the following sentences in the revised paper: "*We identified calibration parameters for each LSMs based on prior studies (Cai et al., 2014; Hogue et al., 2005; andNijssen et al., 2001) and by performing sensitivity analysis. We used soil depths also as calibration parameters following the success of calibrating them in the VIC model(Nijssen et al., 2001; H. Shah and Mishra, 2016). The calibration parameters were manually adjusted so as to match observed streamflow (see Table S2 in Supplementary material).*" on page 4, lines 5-7.

[Figure]

Figure 1: Soil layer thickness (m) for first three layers of the VIC (a,b,c), NOAH (d,e,f) and CLM (g,h,i).

[Figure]

Figure 2: Comparison of persistence before and after calibration. (a,d,g) show change in total depth of first two soil layer after calibration. Panels (b,e,h) show persistence in 60 cm soil moisture before calibration; and (c,f,i) the same but after the model calibration.

(d) It is very confused how to calibrate CLM using soil depth layers. More explanations are needed.

Please refer to the above response.

(e) What are soil parameters in Section2.2.2 and 2.2.3? Are they soil textures (types)? Noah and CLM use the soil textures derived from FAO, and VIC uses soil texture derived Harmonized World Soil Moisture Datbase (HWSD). I am wondering how big differences exist between two datasets? It is very well known that different texture has different soil related parameters such as field capacity, wilting point, etc., which leads to different temporal variation.

We appreciate the insightful comment. Our aim was to compare all three LSMs set-up just after calibration. Hence we have used respective soil texture and LULC. In the revised manuscript, we have shown differences in soil texture and available soil water content over the soil column used in the different models in Fig. 3. As a plausibility check, we show in below the difference in simulated soil moisture anomalies from the VIC model using the soil parameters based on the FAO and HWSD (Fig. 4). We do not find any significant difference in simulated soil moisture anomalies based on these two sets of input information. It is worth noting that the underlying (soil-textural) dataset for the HWSD is mostly derived based on the FAO dataset. The different products may use different pedo-transfer functions to derive soil related parameters. Nevertheless the derived (static) soil parameters do not induce significant differences in the temporal behavior of simulated soil moisture anomalies.

[Figure]

Figure 3: Soil textural information used in three LSMs (a,c,e); and the resulting available water capacity for the total soil column (b,d,f)

[Figure]

Figure 4: Impact of change in soil parameters on soil moisture simulated using the VIC model. (a) Comparison of autocorrelation in 60 cm soil moisture for soil parameters due to different soil parameters based on FAO and HWSD for grid cell at 30.125°N and 77.125°E. (b) Same as (a) but for soil moisture monthly anomalies.

(f) Different vegetation type classification datasets are used for different models, which can result in additional uncertainty for soil moisture product as different vegetation type has different root zone (leads to different transpiration even though surface meteorological forcing is the same).

Thanks. As stated above due to the different requirements of different models in terms of different soil and vegetation parameters, we are enforced to use different vegetation type classification datasets. We will note this issue of additional sources of uncertainty due to requirement of different soil and vegetation datasets (MODIS and AVHRR) in the revised manuscript. However, we have evaluated the sensitivity of vegetation parameters derived from the different sources on soil moisture anomalies, and overall results suggest no major differences in the modeled anomalies (as shown below in Fig. 5).

[Figure]

Figure 5: Impact of change in vegetation cover on soil moisture simulated using the Noah model. (a) Shows comparison of autocorrelation in 60 cm soil moisture due to different vegetation parameters based on MODIS and AVHRR. (b) Same as (a) but for soil moisture.

(g) There is only one test in this study – calibrated run. I would like to see the control run/default run (the default parameters are used) and the comparison with the calibrated run. This will demonstrate what benefits you gain from the calibration process.

This is a good suggestion. We however would like to note that the model calibration may not contribute significantly as we use the anomalies of soil moisture, rather than their absolute values, for drought assessment, which to a large extent is driven by the climate variations.

As suggested by the reviewer, we have performed analysis showing the improvement in models capability to capture stream flow at gauged station after calibration (Table S1). We also show results of similar comparative analysis for the simulated soil moisture (before and after model calibration) below in Fig.s6 and 7. We found insignificant change in a model skill for capturing the temporal dynamics of the simulated soil moisture after calibration at the majority of the investigated locations. In absolute terms, we do see some gain of the calibration –for example the bias in soil moisture fraction (volumetric term) was reduced after calibration as shown in Fig. 7 (below) at IIT Kanpur station.

[Figure]

Figure 6: Correlation of weekly 60 cm simulated soil moisture with the IMD gauge based soil moisture (approx. 60 cm) during the monsoon season for the period 2009-2013. (a-c) Correlation coefficient estimated for the control (default or uncalibrated) set-up for the VIC, Noah, and CLM, respectively. (d-f) Shows change in the correlation coefficient after calibration.

[Figure]

Figure 7: Weekly dynamics of 60 cm simulated soil moisture compared with observations from the IIT Kanpur station. Black line shows the observed soil moisture, while the blue, green and red lines show the simulated soil moisture from the VIC, Noah, and CLM, respectively under the control (un-calibrated) condition; whereas light blue, light green and pink color lines depict the simulated soil moisture after calibration from the respective models.

2. Model evaluation

(a) Calibrated model is only evaluated against observed streamflow. Unfortunately, I am very disappointed that the soil moisture used in this study is not evaluated against either in-situ observations or remotely sensed soil moisture. There are a few stations in India to measure soil moisture from different datasets such as Global Soil Moisture Data Bank (Robock et al. 2000), In-situ observations of soil moisture from India Meteorological Department (Unnikrishnan et al. 2016), and international soil moisture network (https://ismn.geo.tuwien.ac.at/). In addition, quite a few of remotely sensed soil moisture products such as SMAP, SMOS, SMOPS, ASCAT, AMSR2 and more are not used to evaluate LSMs soil moisture simulation products. However, the major variable used in this study is 60 cm soil moisture. Robock, A., et al., 2000: The Global Soil Moisture Data Bank, BAMS, 81, 1281-1299. Unnikrishnan, C. K., et al., 2016: Validation of two gridded soil moisture products over India with in-situ observations, J. Earth System Science, 125, 935-944.

As discussed above, in the revised manuscript we have now compared the simulated soil moisture dynamics with observations available from the IMD stations (Fig. 6) and the Global Soil Moisture Data Bank (Fig. 7). These analyses were conducted for the root-zone soil moisture (taken here as 60 cm of the soil depth). Beside these we also compared the top layer (10-30 cm) simulated soil moisture with the remotely sensed (top few cm) soil moisture from ESACCI (Fig. 8). We however would like to make a note here that most of the gridded (remotely sensed) soil moisture products have their inherent uncertainty – a quite prominent one is that they are limited in their inference of soil water to only few cm from ground surface).

Based on results of these analyses, we have included the following sentences in the revised manuscript on pages 7-8: *"Next we evaluated the skill of the each model for capturing the observed dynamics of near*

*surface and root-zone soil moisture (Fig. S3-S5).We used three different sources of soil moisture observations for this comparison purpose. The first set consisted of the weekly soil moisture observations taken at 18 IMD-based stations during the monsoon season for the period 2009-2013 (Unnikrishnan et al., 2013). The model simulated 60 cm soil moisture dynamics were compared against observations, which generally revealed a good skill for all three models (Fig. S3). Model simulated soil moisture showed a relatively higher correlation with observations in northern and western region as compared to those located in the southern coastal belt.  Among models, the Noah simulated soil moisture exhibited higher correlation as compared to other two LSMs. For this set-up, we also compared the calibrated vs. un-calibrated model runs to understand what improvements (if any) could be achieved by the parameter calibration.  We find limited benefits of the model calibration in this case – only the VIC model benefited by the model calibration mainly in the northern region locations and few of southern locations.*

*The second set of evaluation considered the continuous soil moisture observation datasets at an IIT Kanpur site available from the International Soil Moisture Network (ISMN: Dorigo et al. (2011)). Although all three models exhibited a general bias in capturing absolute values of the observed soil moisture, their daily variability observed over the course of the year is well captured by all three models (Fig. S3). Moreover, the Noah and the CLM models show improvements in terms of reducing overall bias as a result of model calibration.*

*Finally, our third set of model evaluation considered an assessment of the model skill for capturing the remote sensing based soil moisture available from the ESA-CCI product (Dorigo et al, 2012). Here we used the modeled top-layer (10-30 cm) soil moisture simulations over the period 1979-2012 for the comparison (Fig. S4).  Despite the limitation that the ESA-CCI soil moisture inference is for the top few cm of earth surface, we find a positive correlation with modeled soil moisture for all three models across a large part of India. A relatively higher correlation (more than 0.6) can be noticed for regions in the northwest and southern peninsular part of India."*

[Figure]

Figure 8: Correlation of the annual top-layer soil moisture simulated using three LSMs and their ensemble mean (ENS) with a top few cm soil moisture derived from ESACCI during the period 1979-2012.

(b) The authors assumed 60 cm soil layer as root zone. However, for each individual model, it defines its root zone varying from vegetation type to vegetation type. For example in Noah, grass root zone is 1m and forest is 2m. I suggest the authors use 60 cm soil moisture in whole text to avoid confusing the readers.

Thank you. We have revised the text following your suggestion.

3. Model result analysis

(a) The uncertainty analysis is very limited due to three models as the samples are too few for a representative of model uncertainties. In general, the spread can roughly show an uncertainty range when three-model ensemble is used. The authors need to indicate this weakness in a discussion section.

Thanks for your valuable suggestion. We have included following limitation in the revised manuscript on page 13, lines 29-30, "*With respect to the LSMs, we would like to note that the drought uncertainty assessments conducted here are limited to only three LSMs, which is comparatively a smaller size.*"

(b) The authors indicated that the uncertainty in soil moisture is mainly due to model parameterizations – resulting in different persistence of soil moisture. They assumed that there is a large field capacity for CLM but there is no further investigation. In practical, different soil texture datasets, different vegetation type classification datasets, different model structure (specific soil layer in CLM and Noah vs hydrological soil layer concept), and other ET parameterizations may affect this uncertainty together. I recommend make several sensitivity tests to clarify these issues. At least, plot field capacity, wilting point, soil type, vegetation type, root zone depth for all models and then compare their differences.

Our hypothesis for the observed difference in drought characteristics simulated by different model was due to difference in soil moisture persistence – which is inherently driven by soil parameters, especially soil layer thickness. To test this hypothesis, we show in Fig. 9 (below), how the soil moisture persistence differs as a result of differences in the soil layer thickness – before and after the model calibration. Here we can clearly observe the sensitivity of model-simulated values to this key parameter – the higher the soil depth becomes, the larger the persistence is.

[Figure]

Figure 9: Comparison of soil moisture persistence before and after calibration. (a,d,g) Show changes (with respect to default uncalibrated values) in the total soil column depth as a result of the model calibration. Panels (b,e,h) and (c,f,i) show persistence in 60 cm soil moisture before calibration and after calibration for three LSMs, respectively.

(c) In Fig.3c, the seasonal cycles in Noah and VIC are comparable although the magnitude is quite different. However, that in CLM is completely different with Noah and VIC. This further suggests that soil moisture evaluation against in situ observations and remotely sensed product is needed to identify which is closer to the observations.

As mentioned above, we have included an evaluation plot for different LSMs in the revised manuscript (Figs. 6-8). Based on the analysis, we found that the Noah model simulated soil moisture are generally in better agreement to observations as compared to two other LSMs.

(d) In line 33,page 7, the authors cited Wang et al. (2009) to explain higher persistence in soil moisture due to larger water holding capacity and thicker soil column. However, the authors used 60 cm soil layer for all models and also need plot water holding capacity for top 60 cm to verify this point.

We have shown available water in total soil column above and it is very high for CLM based on its highest total soil column depth amongst all models (see Fig. 3 above). Here we would like to make a note that we have shown total column available water, as the water processes will be continuous through all layers.

(e) The authors find an interesting point, that is, there are larger uncertainties in Rabi season than monsoon season. Unfortunately, the authors do not make further investigation to look for the reason. They use a general sentence "which can be associated with the role of air temperature and precipitation on soil moisture" to explain. When Fig. 4 and Fig. S3 are checked, during the monsoon season, three models have larger similarity than Rabi season mainly due to VIC model. A possible reason is that VIC water mode rather than energy mode is used in this study. During the monsoon season, water is unlimited and limited energy is used due to less net radiation (rainy and cloud sky). Energy and water-mode type model does not have big difference. However, during Rabi season, water is limited but energy may be unlimited, so that energy-type model (Noah, CLM) shows larger difference than water-mode type model (VIC). A quick check is to use VIC energy mode to re-run this test to compare with VIC water mode run.

Thank you for the comment. As suggested we run the VIC model in both the water and the water plus energy modes. We however do not find any significant differences in soil moisture simulations between these two modes. A further investigation into disentangling the different sources of uncertainty in different seasons would certainly be an interesting study in its own, but however this would be out of the scope of current study. The uncertainty in the Rabi season can largely be attributed to differences in the soil moisture persistence in the three land surface models.

Minor Comments:

1. Check Table S1: Surface downward shortwave and longwave radiation, for CLMv3.0, soil texture based on IGBP or FAO or vegetation type data based on IGBP.

We have edited this in the revised manuscript.

2.Check Table S2: East coast, calibration and validation period is overlapped.

We have edited this in the revised manuscript.

3. Check Table S2: Mahanadi, calibration and validation period is overlapped.

We have edited this in the revised manuscript.

4. Check TableS2: Subarmarekha, calibration and validation period is overlapped.

We have edited this in the revised manuscript.

I assumed that the authors used independent period to validate the calibrated models. If not, please explain the reason.

Yes, we have used the independent period for the model calibration and validation. We made this point clear in the revised manuscript.

**Response to Reviewer #2 comments**

In this study, three models were implemented to conduct watershed simulation in India. The amazing thing is that the whole India was included, however, quite a few modeling details were missing. Therefore, I probably cannot proceed detailed review at this point. I would suggest adding those details as supplementary information in the next round. On the other hand, there are other similar work done by using multiple models (not limited: Scavia et al. (2017) Sharifi et al. (2017). You did not mention the advantages/disadvantages by using multiple models (and, why these three models???). It cannot always only for the good reasons right? Overall, the content of the given manuscript is way less than it should be (in all sections). Good luck in the next round.

- Scavia, D., M. Kalcic, R. L. Muenich, J. Read, N. Aloysius, I. Bertani, C. Boles, R.Confessor, J. DePinto, M. Gildow, J. Martin, T. Redder, S. Sowa, Y.Wang, H. Yen, 2017.Multiple SWAT models guide strategies for agricultural nutrient reductions. Frontiers inEcology and the Environment, 15(3), pp. 126-132.

- Sharifi, A., H. Yen, K. M. B Boomer,L. Kalin, X. Li, D. E.Weller, 2017. Using multiple watershed models to assess the waterquality impacts of alternate land development scenarios for a small community. Catena,150C, pp. 87-99.

We thank the reviewer for his/her insightful comments. We have included more details in Introduction, Method, Results and discussion, which stress on importance of multi model and its advantages and limitations.

**Response to Reviewer #3 comments**

This manuscript demonstrate the method to reconstruct meteorological and soil moisture droughts in India by three LSMs, i.e. VIC, Noah, and CLM. The overall scientific idea is clearly expressed in detail. The manuscript could be considered to be published after the following minor concerns are addressed. And the language should be carefully polished and make the whole manuscript concise and precise.

We thank the reviewer for his/her insightful comments. We have addressed the reviewer's comments and we will further check the manuscript for language and conciseness.

Specific comments:

1. Page 1 Line 8. In Abstract, "As a large population of India is dependent on agriculture, soil moisture droughts adversely affect agriculture and groundwater resources" This sentence is illogical and should be rephrased.

Thank you. We have changed above sentence to: "*As a large population of India is dependent on agriculture, soil moisture drought affecting agricultural activities (crop yields) have significant impacts on socio-economic conditions*". It is noted on page 1, lines 8-10 in the revised manuscript.

2. Page 5 Line 5. The definition and the formula of SPI and SSI should be clearly expressedin the manuscript instead of just giving the cited literature and leaving the readers tothe literature. In another word, the manuscript should be self-contained.

We have explicitly mentioned the definition (and formula) for SPI and SSI in the revised manuscript in the Supplemental Information.

3. Page 5 Line 18. It would be better if the Indo-Gangetic Plain Region can be shown in supplemental Fig.

Thank you. We now show the Indo-Gangetic Plain region by a green color box in the supplemental Fig. S1.

4. Page 6 Line 20. How do you compare the TWS (which is the total terrestrial water storage including both surface water, soil moisture and groundwater) and the total column soil moisture (which is just the soil moisture stored in aquifers)?

We used the anomaly of the entire soil-water column for every model and compared it against the GRACE derived TWS anomaly. In doing so, we agree with the reviewer that we might have missed the groundwater component (from the VIC simulations); and this might be relevant for a certain part of India. On the other hand, considering that the temporal dynamics of monthly groundwater is rather very slow, the most of the temporal variability in the TWS anomaly can be expressed by the soil-water part. Therefore, comparing the anomaly of the modeled soil water column with GRACE derived TWS anomaly for assessing the skill in terms of capturing the temporal variability, as considered in this study, would be reasonable as shown in Livneh and Lettenmaier (2012). Here, our aim is not to predict TWS using soil moisture, instead we just considered TWS as surrogate of observations to evaluate the models' skill for soil moisture simulation.

5.Page 7 Line 5. Why the ranks of 2014 and 2015 droughts identified in this manuscript are different from the cited literature Mishra et al., 2016b. Which study is confirmed by the in-situ records and which is

biased identification? This affects the quality of the drought reconstruction in this manuscript and should be addressed carefully.

Here we think there is some misunderstanding. In the cited paper Mishra et al, 2016b; we looked for the multi-year drought ranking, rather than a single year drought (as done in this study). Moreover, here the ranking is based on soil moisture drought rather than meteorological drought as in Mishra et al. (2016). Soil moisture droughts are affected by the soil moisture persistence, therefore, ranking (from soil moisture and precipitation) may be different.

6. Page8 Line 2-5. What do you mean by "soil depths were calibrated in all the three LSMs"? Are the soil depths in LSMs the same or not? If they are the same, "The 1-month lag between peak precipitation and peak root-zone soil moisture from the CLM can be due to a relatively deeper soil column." Should be deleted. If not, "since: : :: : :" should be deleted. Above all, you should clearly express the reason and not confuse with each other: soil depths, the number of soil layers or processes related to soil hydrology.

We followed the (calibration) approach of the VIC model in which the soil depths are estimated via a calibration procedure such that the modeled stream-flow matches the observed values. We followed this approach given the wide success of the VIC model application in a wide variety of river basins across different climatic conditions. Moreover using a similar (calibration) approach, we aim to harmonize the different model applications over India. So, the soil depths in LSMs are treated as calibration parameters, which vary across the models and the river basins. We made this point clear in the revised manuscript.

We have mentioned following lines in the revised manuscript, "We used 60 cm depth to represent root-zone soil moisture so as to reduce uncertainty due to different root-zone depth  (based on respective vegetation parameters) and soil layer thickness in three LSMs" on page 2, lines 25-26 and "*We used soil thickness also as calibration parameters following the success of calibrating the VIC using soil layers thickness (Nijssen et al., 2001; Shah and Mishra, 2016)*." on page 4, lines 9-10.

Technical corrections:

1. Page 1 Line 25. The citation Mishra et al., 2016b comes first before 2016a. The literature should be cited in order of their appearance in manuscript.

Thank you. We have revised citation and references.

2. Page 3 Line 4. Digital elevation map (DEM) should be rewritten as digital elevation model (DEM).

Thank you. We have now edited this wording.

**Response to Reviewer #4 comments**

This study reconstructed past droughts over India using multiple land surface models (LSMs). Standardized Precipitation Index (SPI) and Standardized Soil moisture Index (SSI) were used for detection and characterization of meteorological and agricultural drought, respectively. In this study, root-zone soil moisture was estimated from VIC, Noah, and CLM. The parameters of each LSM were calibrated. This study found that there are larger uncertainties in agricultural droughts over a large part of India during crop growing seasons than during monsoon seasons. This study concluded that different persistence of soil moisture from the three LSMs are caused by the difference in model parameterization. Overall, the manuscript is written well but some words and sentences are necessarily revised due to misuses and grammatical errors. The topic is a good-fit to Hydrology and Earth System Sciences (HESS), but I have several major comments on the method and findings. Also, there are several minor comments on the scientific representations, especially Fig.s. More details of the major comments are listed below. Due to the major issues, the current version of the manuscript is not publishable in the HESS. Therefore, I recommend major revision.

We thank the reviewer for his/her insightful comments. We have addressed the reviewer's comments in the revised manuscript.

General Major Comments:

1. It has been very popular to compare the estimated hydro-climate variables from different climate or land surface models (e.g. CMIP3 and CMIP5). One of the lessons from the previous inter-comparison studies is that it is hard to understand what really happens in the models (more likely a black box) unless common parameters (e.g., infiltration capacity or vegetation fraction) across the models and their impacts on the interest estimate (herein, root zone soil moisture (down to 60 cm) are evaluated. In this study, there is a missing section for evaluations of simulated soil moisture, before converting soil moisture to SSI, which give valuable information for how different soil moisture dynamics are across the models. Also, there is a missing for comparisons of the common parameters, which can bring a fundamental understanding of the sensitivity of root-zone soil moisture to the common parameters even though this study discussed that soil water holding capacity (a common parameter) plays an important role in soil moisture dynamics. Therefore, adding sections for root-zone soil moisture analysis and parameter comparison is strongly recommended.

Thank you. As discussed in comments to reviewer #1, we have performed a detailed analysis – evaluating the model skill for soil moisture simulations in three ways. Based on results of these analyses, we have included the following sentences in the revised manuscript on pages 7-8:

*"We evaluated the skill of the each model for capturing the observed dynamics of near surface and root-zone soil moisture (Fig. S3-S5). We used three different sources of soil moisture observations for this comparison purpose. The first set consisted of the weekly soil moisture observations taken at 18 IMD-based stations during the monsoon season for the period 2009-2013 (Unnikrishnan et al., 2013). The model simulated 60 cm soil moisture dynamics were compared against observations, which generally revealed a good skill for all three models (Fig. S3). Model simulated soil moisture showed a relatively higher correlation with observations in northern and western region as compared to those located in the southern coastal belt.  Among models, the Noah simulated soil moisture exhibited higher correlation as compared to other two LSMs. For this set-up, we also compared the calibrated vs. un-calibrated model runs to understand what improvements (if any) could be achieved by the parameter calibration.  We find limited benefits of the model calibration in this case – only the VIC model benefited by the model calibration mainly in the northern region locations and few of southern locations.*

*The second set of evaluation considered the continuous soil moisture observation datasets at an IIT Kanpur site available from the Global International Soil Moisture Network (ISMN: Dorigo et al. (2011)). Although all three models exhibited a general bias in capturing absolute values of the observed soil moisture, their daily variability observed over the course of the year is well captured by all three models (Fig. S3). Moreover, the Noah and the CLM models show improvements in terms of reducing overall bias as a result of model calibration.*

*Finally, our third set of model evaluation considered an assessment of the model skill for capturing the remote sensing based soil moisture available from the ESA-CCI product (Dorigo et al, 2012). Here we used the modeled top-layer (10-30 cm) soil moisture simulations over the period 1979-2012 for the comparison (Fig. S4). Despite the limitation that the ESA-CCI soil moisture inference is for the top few cm of earth surface, we find a positive correlation with modeled soil moisture for all three models across a large part of India. A relatively higher correlation (more than 0.6) can be noticed for regions in the northwest and southern peninsular part of India."*

2. In addition, the output from three LSMs are not able to provide a full distribution of the root-zone soil moisture estimates due to different model structures and parameters. The method introduced in this study might be appropriate for a sensitivity test of the simulated root-zone soil moisture to different land surface model structures and parameters. In Fig. 2, the spreads of areal extents from three models were represented as the envelope but they are actual three points in each year. Or, the authors need to clarify the definition of uncertainty.

We agree with the reviewer on the aspect that three chosen LSMs may not cover the full uncertainty of the root-zone soil moisture estimates. There are number of factors that need to be considered in understanding the full distribution of root-zone soil moisture estimates that include among other things, the uncertainty in forcing variables (precipitation, temperature, etc.), land-surface variables (soil textural information and soil hydraulic parameters like porosity, field capacity and permanent witling point), as well as, model conceptual parameters.

Nevertheless with the application of three models, our aim in this paper was to show the differences in soil-moisture simulations and resulting drought characteristics over India. We term these differences to soil moisture simulations uncertainty to convey the main message that the application of a single model to study soil moisture droughts over India may not be adequate.

Furthermore, we now provide a note in the concluding paragraph of the revised manuscript on future efforts on including more LSMs or hydrological models for analyzing soil moisture drought analysis; as well as for conducting analysis to recognize the contribution from other sources of uncertainty.

Minor comments:

1. Abstract: Page 1 Line 13: "higher uncertainty" should be replaced with "higher sensitivity."

Thank you. Sorry for confusion but we meant here uncertainty amongst drought characteristics estimated using multiple LSMs. We have changed sentence as following: *"We find a higher uncertainty in soil moisture droughts estimated using three LSMs over a large part of India during the major crop growing season (Rabi season, November to February: NDJF) than that of the monsoon season (June to September: JJAS)." It is mentioned on page 1, lines 13-15.*

2. Page 1 Line 18: "multi-model ensemble" should be replaced with "multi-model average." The ensemble is often used for different perturbed physics, initial condition, and forcing within one model.

Thank you. We have incorporated your suggestion in the revised manuscript.

3. Page 1 Line 23: "severity" should be replaced with "intensity" for consistency with the later section.

Thank you. We have revised the word as suggested.

4. Page 2 Line 29-30: What are the temporal coverage of precipitation from 6995 gage stations from IMD? Have the IMD precipitation products compared with the CRU and GPCC (even though they are 0.5 degree)? It is worth to understand how large the uncertainties in precipitation from different sources are.

A detail description of the methodology and underlying dataset used to create the gridded IMD field is provided Pai et al (2014). In this study, we restricted to use the IMD dataset, which uses much more underlying station dataset than used in CRU or GPCC. However, 6995 stations do not have a similar temporal coverage, which to some degree would contribute to an additional source of uncertainty. Nevertheless as we also stated earlier, our goal here is to understand the uncertainty in soil moisture due to usage of different LSMs; we restrict ourselves to use a single set of forcing dataset for all three LSMs.

As a side note, in our recent study, we have compared the IMD based precipitation product with the CRU ones for drought analysis (see Mishra et al., 2016). On a long-time period the two products agree on showing similar basic features, but they do differ (sometime substantially) on a short-time period (see a recent paper by Jing and Wang (2017) in Nature climate change).

5. Page 4 Line 15-18: Zilintikevich coefficient and its explanation should be placed at the end of the sentence.

We have now revised the sentence as suggested. It now reads as: "*We calibrated the Noah model parameters that include depth of four soil layers, Zilintikevich coefficient surface runoff parameter, and bare soil evaporation component. Zilintikevich coefficient controls the ratio of the roughness length for heat to the roughness length for the momentum, and thus representing an aerodynamic resistance term.*" Please find it on page 5, lines 12-14 in the revised manuscript.

6. Page 5 Line 10-11: Is a Gamma (parametric) distribution appropriate in computing a agricultural (soil moisture) drought index? What about using percentiles (nonparametric) as a drought index?

We have tested the appropriateness of the distribution function. We used here the parametric (Gamma) form for the agricultural drought so to be consistent with the precipitation based drought index (SPI). We, however, agree with the reviewer of using a more robust non-parametric based drought index (like percentile) – we adopted for such approach in the recent past (see Mishra et al, 2016b) – but here in this study we do not find big differences in our modeling results due to different approach of estimating drought indices (the main results and conclusions however remain unaffected).

7. Page 5 Line 17: Why this study uses the 4-month SSI? I assume that it was matched with Rabi seasons but there is no explanation about it. Please clarify it.

Yes, the reviewer is right in his/her interpretation. We added a sentence to clarify this in the revised manuscript.

8. Page 5 Line 20: Drought severity is defined as the total area (intensity x duration) from initiation through recovery. "for each year, mean severity of droughts" is confusing. Please change the sentence as "for each year, mean 4-month SSI value was ..."

Thank you. We have reformulated this sentence in the revise manuscript as suggested.

9. Page 6 Line 7: "the ensemble mean streamflow" should be replaced with "the multi-model averaged streamflow."

Thank you. We have reformulated this wording in the revise manuscript as suggested.

10. Page 7 Line 11-12: What is the definition of uncertainty in area extent of drought? Please clarify it either here or in the method section.

For this analysis, we have taken one standard deviation of simulated areal drought extent of different LSMs to represent the uncertainty.

11. Page 8 Line 32-33: How can higher persistence of CLM soil moisture can be attributed to its higher water holding capacity and thicker soil column? Please explain the possible physical processes. The explanation will be beneficial for readers.

We attribute the higher persistence of the CLM soil moisture to a relatively thicker soil column that acts as a damping factor to meteorological forcing. This is seen in seasonal cycle of 60 cm soil moisture from all three models (Fig. 3c). The CLM responds slower as compared to the VIC and Noah after the monsoon season. This is verified in supplemental fig S8, where autocorrelation of the CLM is higher as compared to the VIC and Noah in response to monsoon season precipitation.

12. Page 9 Line 21-22: This study finds that regardless of seasons, precipitation is a major driver for drought and temperature is a minor. If then, uncertainties in meteorological forcings, especially precipitation might be more important than uncertainties in soil moisture. Why didn't this study investigate uncertainties in precipitation?

This is a good point. We would like to reiterate here that the goal of this study is to analyze the uncertainty in soil moisture simulation due to the choice of LSMs. There are different sources of uncertainties – and clearly the uncertainty in climatic forcing (most importantly precipitation) would be more dominant among others; analyzing the uncertainty due to different forcing datasets would be certainly interesting but at this point this would be out of the scope of this study.

13. Page 11 Line 1-15: Please discuss the potential implementation of the findings in the section 3.5.

[revised manuscript text omitted]

Admin 29/11/2017 13:39

Admin 29/11/2017 13:39

Admin 29/11/2017 13:39

Admin 29/11/2017 13:39

Admin 29/11/2017 13:39

Admin 29/11/2017 13:39

Admin 29/11/2017 13:39

Admin 29/11/2017 13:39

Admin 29/11/2017 13:39

Admin 29/11/2017 13:39

Admin 29/11/2017 13:39

Admin 29/11/2017 13:39

Admin 29/11/2017 13:39

Admin 29/11/2017 13:39

Admin 29/11/2017 13:39

Admin 29/11/2017 13:39

Admin 29/11/2017 13:39

Admin 29/11/2017 13:39

Admin 29/11/2017 13:39

[Figure]

Figure 2: Uncertainty in areal extent (%) of 60 cm soil moisture drought simulated using the three
LSMs (i.e. VIC, Noah, and CLM). (a) Multimodel ensemble (brown) mean 4-month Standardized Soil
Moisture Index (SSI) and inter-model variation (shaded) estimated as one standard deviation for the
monsoon season. Black line in (a) shows 4-month Standardized Precipitation Index (SPI) at the end of
the monsoon season (June through September) (b) multimodel ensemble mean and uncertainty in 4-
month SSI estimated using the three LSMs for the Rabi season (November through February). Light
brown shaded area shows uncertainty in severe-to-exceptional drought based on model simulated SSI
(SSI <-1.3). Dark brown line shows areal extent estimated based on ensemble mean SSI for the three
LSMs. Grey line marks top drought years based on area under drought.

Admin 29/11/2017 13:39

Admin 29/11/2017 13:39

[Figure]

Figure 3: Uncertainty in persistence in root-zone soil moisture (60 cm). Seasonal cycle of all-India averaged (a) precipitation (b) mean air temperature and (c) 60 cm soil moisture simulated using the VIC (blue), the Noah (green), and the CLM (red). (d,e,f ) Autocorrelation in 60 cm soil moisture at 4-month lag simulated using the VIC, Noah, and CLM, respectively. (g) All-India median autocorrelation in the 60 cm soil moisture from the three LSMs.

Admin 29/11/2017 13:39

Admin 29/11/2017 13:39

Admin 29/11/2017 13:39

[Figure]

Figure 4: Reconstruction of monsoon season drought events of (a-f) 1987, (g-l) 2002 and (m-r) 2015, estimated based on (a,g,k) 4-month SPI at the end of the monsoon season, (c,i,o) 4-month SSI at the end of the monsoon season simulated using the VIC model, (d,j,p) same as (c,i,o) but for the Noah model, and (e,k,q) same as (c,i,o) but for the CLM. (f,l,r) Ensemble mean 4-month SSI simulated using the VIC, Noah, and CLM. (b,h,n) Air temperature anomaly during the monsoon season for the selected drought years.

[Figure]

Figure 5: Uncertainty in Intensity-Areal Extent-Frequency (IAF) curves for the monsoon season 60 cm soil moisture drought estimated using the three LSMs. Dark brown color shade shows uncertainty in models without considering parameter uncertainty in the Generalized Extreme Value (GEV) distribution while light brown color shows uncertainty considering 95% confidence interval of the GEV parameters for return periods (a) 10, (b) 20, (c) 50, (d) 100, (e) 200, and (f) 500 years. Black error-bars indicate uncertainty for the 2002 monsoon season drought using three LSMs.

Admin 29/11/2017 13:39

[Figure]

Figure 6. (a,c,e) Relationship between monsoon season precipitation anomaly (%) and 4-month SSI at the end of the monsoon season; and (b,d,f) same as (a,c,e) but for the relationship between 4-month SSI and air temperature anomaly of the monsoon season. Correlation coefficients are shown for all-India SSI (blue) and 4-month SSI over the Indo-Gangatic Plain (red).

Admin 29/11/2017 13:39

---

## Author Response (AR2)

February 02, 2018

Editor, Hydrology and Earth System Science

Dear Dr. Franssen,

We are pleased to submit our revised manuscript "**Reconstruction of droughts in India using multiple land surface models (1951-2015)"** by Mishra et al.

We have successfully addressed all the comments and suggestions provided by the reviewers and yourself. We appreciate constructive comments and suggestions that have helped in improving our work.

We provide a detailed response document and changes are highlighted in the revised manuscript. We have added a new figure (Fig. S16) showing the relationship between soil depth and available water (and soil depth and soil moisture persistence) for the three (VIC, Noah, and CLM) land surface models. Overall, we find that the soil depth in the three LSMs is a major driver of soil moisture persistence, which in turn affects drought severity. Since our manuscript focuses on soil moisture drought, the other factors (related to vegetation, soil, and calibration) that may contribute to uncertainty in drought assessment, play a relatively minor role (please see supplemental Figures S2, S3, S4, and S16) for more details.

We thank you once again for your constructive comments on our important work and look forward towards the positive decision.

Sincerely,
Vimal Mishra
Associate Professor
Civil Engineering
Indian Institute of Technology (IIT) Gandhinagar

We thank all the reviewers and the editor for their constructive and insightful comments that have helped in improving the revised version.

5 Editor Decision: Publish subject to revisions (further review by editor and referees) (22 Jan 2018) by Harrie-Jan Hendricks Franssen

Comments to the Author:

Your manuscript "Reconstruction of droughts in India using multiple land surface models (1951-2015)" has been subjected now to re-review by two of the original reviewers. One of reviewers suggests rejection of the manuscript and another

10 reviewer minor revision. Unfortunately, two of the former reviewers declined to review the revised version. I checked the handling of the comments of reviewers #1 and #4 and my additional comments are:

- Comments (1e) and (1f) of reviewer #1 should be more reflected in the manuscript. I suggest to add some additional information, without entering into too many details.

We appreciate your suggestions and we have added more details related for comments 1(e) and 1(f) in the revised

15 manuscript. We have addressed the comment 1(e) in detail, which pertains to the differences in soil texture in the land surface models.

(1e) What are soil parameters in Section2.2.2 and 2.2.3? Are they soil textures (types)? Noah and CLM use the soil textures derived from FAO, and VIC uses soil texture derived Harmonized World Soil Moisture Datbase (HWSD). I am wondering how big differences exist between two datasets? It is very well known that different texture has different soil related

20 parameters such as field capacity, wilting point, etc., which leads to different temporal variation.

Soil parameters are like porosity, field capacity, and wilting point among others. As such soil texture types are not model parameters, but these information are used to derive above mentioned soil parameter using pedo-transfer functions. In the revised manuscript, we show the differences among soil parameters derived based on two different underlying textural datasets. Based on our analysis results, we have added the following text in the revised manuscript:

25 *"However, since that land surface models use soil parameters from the two different sources that can lead to differences in available water (Fig. S2), we evaluated the difference in the simulated soil moisture anomalies from the VIC model using the soil parameters based on the FAO and HWSD datasets (Fig. S3). We do not find a significant difference in the simulated soil moisture anomalies based on the soil parameters from the two sources. It is worth noting that the underlying (soil-textural) dataset for the HWSD is mostly derived based on the FAO dataset. The different products may use different pedo-transfer*

30 *functions to derive soil related parameters. Nevertheless the derived (static) soil parameters do not induce significant differences in the temporal dynamics of simulated soil moisture anomalies (Fig. S2-S3)." (page 4)*

We note here that our comparative analysis is based on soil moisture anomalies (relevant for estimating indicator of droughts), and not the absolute values of simulated soil moisture, which could show (considerable) sensitivity to underlying soil hydraulic parameters.

(1f) Different vegetation type classification datasets are used for different models, which can result in additional uncertainty for soil moisture product as different vegetation type has different root zone (leads to different transpiration even though surface meteorological forcing is the same).

To address this concern, we evaluated the model simulated soil moisture with different vegetation related datasets. Based on the analysis result, we have added the following text in the revised manuscript to address this comment:

*"Vegetation characteristics specified within LSMs were derived from the Advanced Very High-Resolution Radiometer (AVHRR) and Moderate Resolution Imaging Spectroradiometer (MODIS) datasets. We evaluated the sensitivity of vegetation parameters derived from AVHRR and MODIS on simulated soil moisture using Noah model (Fig. S4). We do not find any substantial differences in the dynamics of the simulated soil moisture anomalies due to differences in the underlying vegetation parameters estimated from AVHRR and MODIS datasets (Fig. S4)." (page 4)*

- Comment (3b) of reviewer #1 should be answered, i.e. a comparison of the parameters for the different models, as indicated by the reviewer should be made.

We have evaluated the sensitivity of soil parameters derived based on FAO and HWSD datasets (Fig. S3). Similarly, we have estimated the differences in simulated soil moisture dynamics due to vegetation parameters derived from AVHRR and MODIS (Fig. S4). Both of these analysis resulted in no major differences in simulated soil moisture anomalies. We agree that there can be different sources of uncertainty; however, our main focus in this study was to understand the uncertainty in soil moisture drought assessment due to selection of three land surface models. We find that the differences in available water and soil moisture persistence primarily contribute to the differences in simulated soil moisture drought (as shown in Fig. S2). Moreover, we find that differences in depth of soil layers (Fig. S9) of the three land surface models (VIC, Noah, and CLM) that also contribute to differences in available water, which in turn affects the soil moisture persistence behavior (Fig. S8). Differences in soil moisture persistence further affect the analyzed drought characteristics like areal extent and severity of droughts (Fig. S10).

Please see the response to the next comment for more details.

- Comment 1 of reviewer #4 is only partly handled and the relation between parameter values and states should also be addressed.

In this respect, we have compared (calibrated vs uncalibrated) soil parameters and resulting modeled variables, and found that they do not lead to a significant difference in simulations of soil moisture dynamics (Fig. S8). The major factor affecting soil moisture persistence and resulting drought characteristics is soil depth prescribed in the three selected models (Fig. S9 and S8

We have added a new figure (Fig. S16) to highlight the relationship between soil depth and available water, as well as between soil depth and soil moisture persistence. We have added the following text in the revised manuscript:

*"We found that persistence in soil moisture is strongly linked with soil layer thickness (see Fig. S8), which in-turn affects the soil water holding capacity (available water, Fig. S16). Apart from the soil moisture persistence, there can be several other*

*factors that can introduce uncertainty in 60 cm soil moisture simulations. For instance, all the three LSMs have different calibration parameters, which do not cover the entire range of uncertainty due to manual calibration (De Lannoy et al., 2006; Samaniego et al, 2013). However, our results show that the model calibration has a little impact on soil moisture anomalies which are largely driven by the climate forcing."* (page 14)*"*

[Figure]

Fig. S16. Relationship between soil depth and all- India median available water (a), and soil depth and all- India median soil moisture persistence (b) from the three (VIC, Noah, and CLM) land surface models.

I suggest to handle the remaining comments listed above as well as the comments of the two reviewers in a new, revised version. I believe that the remaining comments can be handled by a moderate revision. I am looking forward to the revised version of your paper.

Thanks. We appreciate your suggestions and comments, which have been helpful in improving the quality of our work.

**Anonymous Referee #2**

Suggestions for revision or reasons for rejection (will be published if the paper is accepted for final publication)

I thought I was giving a fairly friendly comments in the first round. However, I do not see my comments being properly addressed by authors. My comments were so simple: "You did not mention the advantages/disadvantages by using multiple models (and, why these three models???)". There are reasons to apply multi-model approach, what about yours?

We are bit disappointed that the reviewer did not find our revision efforts adequate enough. We would like to bring the reviewer attention to the second and third paragraphs of Introduction in the revised manuscript, which highlight the need of using multiple models for drought assessment.

**Report #2**

The authors have clearly addressed all my previous comments. However, I still have a few minor concerns and suggestions.

Thanks. We appreciate your insightful suggestions and comments.

1. The title of this manuscript is about drought reconstruction. However, the abstract and conclusions seem to focus more on the uncertainties among the three land surface models (LSMs). It is suggested to add a few key points about the reconstructed droughts in sections of Abstract and Conclusions.

Thanks. We have modified the abstract and conclusions to include the key points of the reconstructed droughts. Specifically, we have added the following text in the revised manuscript:

*"Based on simulations from the three LSMs, we find that major drought events occurred in 1987, 2002, and 2015 during the monsoon season (June through September). During the Rabi season (November through February), major soil moisture droughts occurred in 1966, 1973, 2001, and 2003. Soil moisture droughts estimated from the three LSMs are comparable in terms of their spatial coverage, however, differences are found in drought severity." (page 1, Abstract)*

2. In the manuscript, the author primarily focused on the 60cm-depth soil moisture drought. Is there any evidence for the selected threshold (60 cm)? It is suggested to add a sensitive analysis or a few sentences to discuss whether the varying threshold of soil depth conclusions may affect the final conclusions.

We selected the top 60 cm to estimate soil moisture–based drought because, for many crops, effective root zone depth falls in this region (Jalota and Arora 2002).

We have added the following text in the revised manuscript:

*"We selected the top 60 cm to analyze soil moisture drought because, for many crops, effective root zone depth falls in this region (Jalota and Arora 2002)." (page 2)*

3. Section 2.4 (Line 18-19). Please provide the climatoloigcal period of the monsoon and Rabi seasons, respectively.

Thanks, we have mentioned that the climatological period is 1951-2015. (page 7)

4. Figure 1. If I understand correctly, the performance of VIC model, after calibration, even seems worse than before calibration for a few stations. If so, why we have to do calibration? Please clarify it in the revised version.

Thanks. After calibration, soil moisture simulations improve at few stations and worsen at few other ones; latter was especially the case located in the peninsular India. Our primary variable used for calibration was streamflow as soil moisture data (long-term) was not available for all the river basins in India. In this respect, the evaluation of model against soil moisture constitutes an independent test case.

[revised manuscript text omitted]